# From Harm to Help: Turning Reasoning In-Context Demos into Assets for Reasoning LMs

## Abstract

Recent reasoning LLMs (RLMs), especially those trained with verifier-based reinforcement learning, often perform worse with few-shot CoT than with direct answering. We revisit this paradox using high-quality reasoning traces from DeepSeek-R1 as demonstrations and find that adding more exemplars consistently degrades accuracy, even when demonstrations are optimal. A detailed analysis reveals two mechanisms behind this decline: (i) *semantic misguidance*, where high textual similarity leads the model to treat the target as the same as the exemplar and to copy intermediate steps verbatim; and (ii) *strategy transfer failure*, where the model struggles to extract useful reasoning strategies and apply them to target questions. Guided by these, we introduce Insight-to-Solve (I2S), a sequential test-time procedure that turns demonstrations into explicit, reusable insights and derives a target-specific reasoning trace; optionally, the reasoning is self-refined for coherence and correctness (I2S⁺). Extensive experiments on diverse benchmarks show that I2S and I2S⁺ consistently outperform both direct answering and test-time scaling baselines across open- and closed-source models. Even for GPT models, our method helps: on AIME'25, GPT-4.1 rises by $+14.0\%$, and o1-mini improves by $+2.7\%$ on AIME and $+1.7\%$ on GPQA, indicating that in-context demonstrations can be harnessed effectively via insight–refine–solve framework.

## 1 Introduction

Large language models (LLMs) exhibit emergent in-context learning (ICL) capabilities, solving diverse tasks by conditioning on a few input-output exemplars without parameter updates (Brown et al., 2020). Chain-of-Thought (CoT) prompting (Wei et al., 2022) further augments this paradigm by incorporating step-by-step demonstrations, guiding models to decompose complex problems into intermediate steps. This approach, commonly referred to as Few-shot CoT, has become the de facto standard for enhancing reasoning during evaluation across numerous benchmarks, from mathematical problem-solving to commonsense reasoning (Cobbe et al., 2021; Hendrycks et al., 2021; Contributors, 2023; Huang et al., 2023; Wang et al., 2024).

However, recent reasoning LLMs (RLMs) (Guo et al., 2025; OpenAI, 2024; Yang et al., 2025), especially those trained with reinforcement learning with verifier rewards (RLVR) (Zheng et al., 2025a; Yu et al., 2025; Liu et al., 2025b; Zhou et al., 2025), reveal a counterintuitive phenomenon: few-shot CoT can *hurt* performance. DeepSeek has expressed concern about the sensitivity of RLMs to prompts and that the standard few-shot CoT prompts used in benchmarks may actually hurt RLMs performance, leading them to adopt direct inference without few-shot demonstrations in evaluations (Guo et al., 2025). This concern is echoed in OpenAI's cookbook for their reasoning models, e.g. GPT O-series, which explicitly recommend avoiding CoT prompts (OpenAI). Across open-source RLMs, several empirical observations likewise find direct answers (zero-shot) outperform carefully crafted few-shot CoT prompts (Zheng et al., 2025b). A plausible explanation is that legacy CoT demonstrations in current benchmarks (Cobbe et al., 2021; Huang et al., 2023; Wang et al., 2024) were cooked for earlier, weaker models and now lag behind the reasoning traces that can be generated by RLMs on their own, thereby constraining rather than enhancing their capabilities (Cheng et al., 2025). This naturally raises a question: *With high-quality reasoning demos, can few-shot prompting help RLMs?*

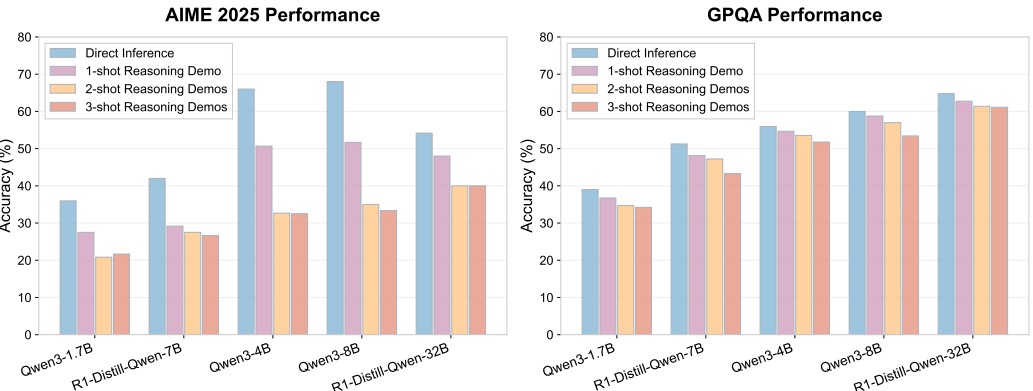

Figure 1: Few-shot CoT harms the RLMs performance, even with high-quality reasoning demonstrations.

To test this, we condition RLMs on long, high-quality reasoning traces generated by DeepSeek-R1 (Guo et al., 2025) for problems closely matched to each target question. We then evaluate DeepSeek-distilled Qwen and Qwen3 series models on the target questions. As shown in Figure 1, even with such demonstrations, accuracy consistently drops relative to direct answer, and the decline becomes more pronounced as more reasoning exemplars are added. To understand why demonstrations whose solution procedures already mirror the target are helpful to humans but harmful to RLMs, we conduct in-depth study, which reveals two key mechanisms behind this counterintuitive effect: (1) **Semantic misleading**: Reasoning traces from similar questions should help by illustrating strategy, but high textual similarity biases the model to treat the target as the *same* problem, overriding its distinct logical structure and causing near-verbatim copying of intermediate steps and answers; (2) **Strategy transfer failure**: RLMs struggle to *extract* beneficial reasoning strategies from demonstrations and *apply* to target questions, even when those demonstrations include optimal solution paths. The models' internal reasoning mechanisms for the target question appear to conflicts with their ability to leverage external reasoning examples.

Guided by these findings, we introduce **I2S** (Insight-to-Solve), a lightweight sequential test-time scaling method that enhances the in-context learning of RLMs. The method converts demonstrations into actionable guidance while avoiding semantic copying. Concretely, we place demonstrations alongside the target question, extract reusable insights, and apply them to derive a reasoning trace for the target. Optionally, we let the model perform a brief self-refinement, conducting consistency checks, identifying potential issues, and editing the trace into a smooth, target question aligned reasoning; we denote this variant **I2S⁺**. Finally, we perform *decoupled solution generation* by conditioning solely on the target question and the (optionally refined) reasoning trace, explicitly preventing reproduction of any demonstration content. This Insight–Refine–Solve design tackles the two failure modes head-on: explicit insight extraction mitigates *strategy-extraction failure*, meanwhile decoupled solving reduces *semantic misleading*.

Empirically, I2S and I2S⁺ deliver consistent gains across open- and closed-source models. On AIME'25/GPQA (close-ended benchmarks) with DeepSeek-R1-Distill-Qwen-7B, I2S⁺ improves over direct answering by +12.0/+4.8; with Qwen3-1.7B, by +6.7/+3.5 (Table 1). Closed-source models also benefit: on GPT-4.1, I2S raises AIME from 34.0 to 48.0 (+14.0) with a modest GPQA gain; o1-mini improves by +2.7 (AIME) and +1.7 (GPQA) (Figure 2). On the open-ended *General Reasoning* tasks (with GPT-4.1 as judge), I2S and I2S⁺ outperform direct answering on both Engineer and General questions with gains up to +2.1 across closed-source models (Table 2). We summarize our contributions as follows:

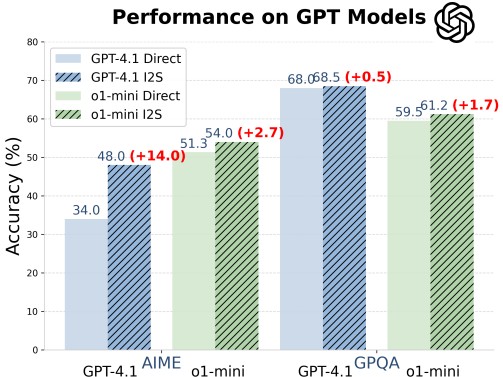

Figure 2: Performance improved across powerful closed-source models such as GPT-4.1 and o1-mini.

- **Revisiting reasoning demonstrations for RLMs.** Even high-quality, closely matched demos can *hurt* RLMs; we identify two causes: *semantic misleading* (treating similar problems as identical) and *strategy-extraction failure* (failing to transfer the solution strategy).

- **I2S: turning demos into assets at test time.** A test-time pipeline that extracts transferable insights, optionally self-refines a target question aligned trace (I2S$^+$), and decouples *solution generation* from demos to prevent copying.
- **Consistent gains across models and benchmarkes, including GPT.** Across open and closed models, I2S/I2S$^+$ yield consistent gains on AIME'25, GPQA, and open-ended task General Reasoning, exceeding baselines without extra training or heavy sampling.

## 2 RELATED WORK

**Few-Shot CoT Prompting.** Chain-of-Thought (CoT) prompting, which elicits step-by-step reasoning by providing a few demonstrations in the prompt, is a cornerstone of in-context learning (ICL) for complex tasks (Brown et al., 2020; Wei et al., 2022). This has led to extensive work on automating the selection and generation of reasoning exemplars (Li et al., 2023; Qin et al., 2023). Providing detailed reasoning demonstrations became a standard approach for complex reasoning tasks, with follow-up work focusing on automating the retrieval, selection and generation of exemplars (Li et al., 2023; Qin et al., 2023; Peng et al., 2024; Sevinc & Gumus, 2024). However, the efficacy of this approach is now being questioned. Recent studies show that CoT can be detrimental for certain tasks (Zheng et al., 2025b) and that for powerful models, it may merely enforce output formatting rather than improve reasoning (Cheng et al., 2025). The emergence of reasoning-specialized LLMs (RLMs) further complicates the ICL landscape. RLMs, often trained with reinforcement learning, generate high-quality and detailed reasoning traces. The few-shot CoT demonstrations commonly used in benchmarks were crafted for earlier, weaker models and may now constrain rather than guide an RLM's reasoning. While some work on RLMs has abandoned demonstrations (Guo et al., 2025), we investigate whether their effectiveness can be restored by using higher-quality reasoning traces sourced from even stronger models.

**Test-Time Scaling.** Test-Time Scaling (TTS) aims to enhance model performance by dedicating more computational resources during inference (Brown et al., 2024; Wu et al., 2025; Snell et al., 2025). Current methods fall into two main categories: *parallel* and *sequential* scaling. Parallel scaling involves generating multiple candidate outputs and aggregating them using a specific strategy (Brown et al., 2024; Zeng et al., 2025; Stroebl et al., 2024; Sun et al., 2024; Gui et al., 2024; Snell et al., 2025; Liu et al., 2025a; Wu et al., 2025; Jiang et al., 2023; Li et al., 2025; Chen et al., 2023a). Aggregation can occur at the solution level, using reward-guided techniques like Best-of-N Search (Sun et al., 2024) or guidance-free methods like Majority Voting (Wang et al., 2022) and Universal Self-Consistency (Chen et al., 2023a). Alternatively, aggregation can be applied at intermediate steps using tree-search algorithms like Beam Search (Qiu et al., 2024; Yu et al., 2023; Kool et al., 2019) and MCTS (Zhang et al., 2023; Chen et al., 2024a), which rely on guidance signals to select optimal paths. Other aggregation strategies further diversify this approach, including difficulty-aware model selection (Snell et al., 2025) and reward-guided voting (Wu et al., 2025). Sequential scaling enhances solutions by iteratively refining them along a single generation path, which can be done through self-revision, where a model evaluates and improves its own output (Madaan et al., 2023), or by leveraging external feedback (Chen et al., 2023b; Gou et al., 2024). While the effectiveness of unguided self-revision is debated, some work shows it is a learnable skill, positing that evaluating a solution is an easier task than generating one (Kumar et al., 2024; Zhang et al., 2024). Empirically, with high-quality feedback, self-revision can outperform parallel methods (Chen et al., 2024b). Moving beyond a simple self-revision sequential scaling paradigm, our work focuses on how to effectively use reasoning demonstrations in the context.

## 3 REVISITING IN-CONTEXT REASONING DEMONSTRATIONS FOR RLMS

We use ***in-context reasoning*** to refer to the prompting way that an RLM, conditioned on a prompt containing long reasoning demonstrations, first generates a reasoning trace and then produces the final answer. To study whether reasoning demonstrations actually help RLMs, we follow a controlled in-context learning setup.

### 3.1 REASONING DEMONSTRATIONS DEGRADE REASONING MODELS

**Datasets and models.** We evaluate models on two challenging reasoning benchmarks: AIME 2025 (Mathematical Association of America, 2025) for mathematical problem solving and GPQA

Diamond (Rein et al., 2024) for logical reasoning problems. For models, we use five models spanning different scales and architectures: Qwen3 models (1.5B, 3B, and 8B parameters) and two DeepSeek-R1-Distilled models (7B and 32B parameters). The DeepSeek-R1-Distilled models are distilled from DeepSeek-R1 and thus have been exposed to reasoning traces of R1 during training. 114k high-quality examples Our demonstration set, OpenThoughts-114k (Guha et al., 2025), contains question–solution pairs covering math, science, code, and puzzles. For each target test question we retrieve the most relevant demonstrations using a retriever based on question embeddings, similar to Query-RAG (Ma et al., 2023), and prepend them to the input. This retrieval ensures that demonstrations are not arbitrary but contribute to the solving of target problem, providing a strong test of whether RLMs can leverage high-quality reasoning examples. We compare this few-shot setup against a direct inference baseline (i.e., zero-shot with no demonstrations).

**Results.** Figure 1 shows that, across *all* models and *both* benchmarks, adding reasoning demos consistently reduces accuracy compared to direct inference, with degradation *worsening* as the number of shots increases. The effect is especially pronounced on *AIME'25* (single-demo drops of roughly 6-16% in accuracy and 3-shot drops up to 35%); and is still non-trivial on *GPQA-Diamond* (1-5% at 1-shot, 4-8% by 3-shot). Besides, larger models are not immune; the 32B distilled model also declines as more demos are introduced. Notably, the DeepSeek-R1–distilled models were trained on DeepSeek-R1–generated traces, i.e., the same style of reasoning we place in context. According to the theoretical view *"LLMs learn in-context via gradient descent"* (Von Oswald et al., 2023; Dai et al., 2022; Ahn et al., 2023), such conditioning should approximate continued training without distribution shift. Yet, even under this seemingly favorable condition, in-context exposure to these traces produces sizable performance drops.

### 3.2 Diagnosing In-Context Reasoning Failures

**Probabilistic Perspective on In-Context Reasoning.** To analyse why demonstrations may backfire, we model test-time reasoning in RLMs as two latent stages: (i) *reasoning generation*, which produces a scratchpad $z$ (e.g., `<think>...</think>`) to form hypotheses, carry out calculations, and self-correct; and (ii) *answer generation*, which maps a target question $x$ with $z$ to the final answer $y$. The joint distribution can thus factorize as

$$p(y, z \mid x) \;=\; p(z \mid x)\, p(y \mid x, z).$$

When a $k$-shot prompt of reasoning demonstrations $\mathcal{D} = \{(x_i, z_i, y_i)\}_{i=1}^{k}$ is provided, where each $z_i$ is an exemplar reasoning trace connecting $x_i$ to $y_i$, both stages condition on $\mathcal{D}$:

$$p(y, z \mid x, \mathcal{D}) \;=\; \underbrace{p(z \mid x, \mathcal{D})}_{\text{reasoning trace}} \underbrace{p(y \mid x, z, \mathcal{D})}_{\text{final answer}}. \tag{1}$$

Eq. 1 factorizes the impact of reasoning demonstrations $\mathcal{D}$ on the reasoning process into two phases: (i) the model's ability to generate a coherent and useful reasoning trace $p(z \mid x, \mathcal{D})$, and (ii) its ability to extract the correct answer given the trace $p(y \mid x, z, \mathcal{D})$. In the following sections, we analyze each component in depth to uncover the mechanisms underlying in-context reasoning failures.

### 3.3 Why Do Reasoning Demonstrations Backfire?

**Semantic Misleading from Demonstration Content.** When the demonstration and target question share high lexical overlap, the model often treats them as equivalent, directly copying intermediate steps and even the final conclusion. This risk is amplified when demonstrations use domain-specific notation or phrasing that superficially resembles the target. As shown in Figure 3 (Left), the shared token "digits 1–8" cues the model to transplant the demo heuristic "split into two groups" and "sum to around 18" (useful for minimizing the difference between two 4-digit numbers). The model then misapplies this surface heuristic to the target, concluding that the *odd-position* and *even-position* sums are both 18. This ignores the target's structural requirements for divisibility by 22: the last digit must be even and the alternating digit sum must be $0 \bmod 11$. The demo acts as a *semantic lure*: it highlights surface cues while obscuring the structure of the problem.

**Failure to Transfer Underlying Reasoning Structure.** Even when demonstrations encode genuinely useful structural insights, models can fail to transfer (*extract* and *apply*) them appropriately.

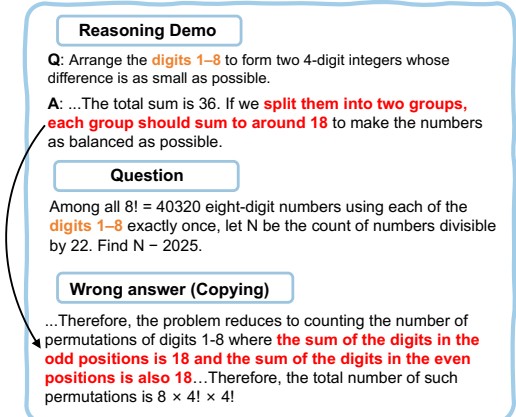
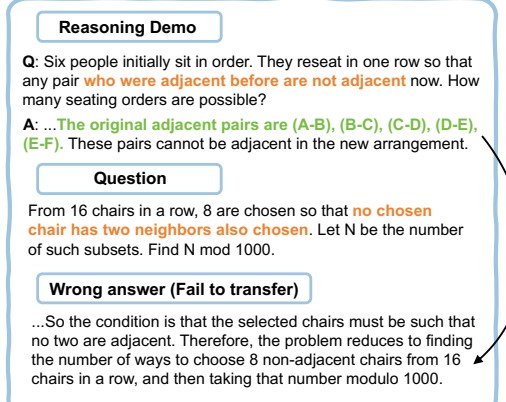

Figure 3: Failure modes in in-context reasoning. Left: Semantic misleading (red: inappropriate copying). Right: Failed structural transfer (green: missed transferable insight).

In Figure 3 (Right), the demo's explicit listing of adjacent pairs $(A-B), (B-C), \ldots$, illustrating the method: turn an abstract rule into explicit forbidden patterns. The target should borrow exactly this approach: enumerate seat adjacencies $(1-2), (2-3), \ldots, (15-16)$ and apply it by recognizing the target constraint: "no chosen chair has two neighbors also chosen". This condition forbids consecutive triplets of chosen chairs. It is fundamentally different from forbidding pairs, used in the demo. Instead, the model *mis-extracts*—skipping adjacency enumeration; and *mis-applies*—defaulting to "no two are adjacent", collapsing a triplet-wise constraint into a pairwise one.

**Mechanistic Discussion.** Taken together, these observations suggest a simple mechanistic picture. Because demonstrations and targets share a single context, the strong cross-attention from target tokens to highly similar tokens of in-context demostrations effectively implements a "short-cut completion": the model continues the most likely demo-like pattern instead of re-deriving the solution from the target constraints. This is consistent with our observations that LLMs can over-weight surface similarity when reading multiple QA pairs in a single prompt (*Semantic Misleading*). At the same time, useful reasoning strategies (e.g., "set up equations before plugging in numbers") are abstract and require inductive transfer rather than copying. When demonstrations are long, the model might focus on local token-level patterns instead of extracting high-level invariants, leading to failure to apply the correct strategy to the target (*Failure to Transfer*).Section We view our discussion as an empirical probe of these inductive biases, and leave a formal theoretical treatment of these mechanisms to future work.

### 3.4 A QUICK FIX: "TWO-STEP" INFERENCE

In in-context reasoning paradigm, both the reasoning trace and the final answer are conditioned on the demonstration set, making both stages vulnerable to *semantic misleading* (e.g., verbatim copying triggered by superficial similarity). A straightforward mitigation is to *decouple* answer generation from reasoning: first elicit a reasoning trace using the demos, then discard the demos and generate the final answer using only the question and the elicited trace. As illustrated in Figure 5 (right), we obtain a trace $z$ from the demo-augmented prompt, cut at the `<think>` boundary, and concatenate $(x, z)$ to produce $y$—thereby shielding the answer stage from the demos. Under the same evaluation protocol of Figure 1, this decoupling yields consistent gains over naive single-pass 1-shot in-context prompting across

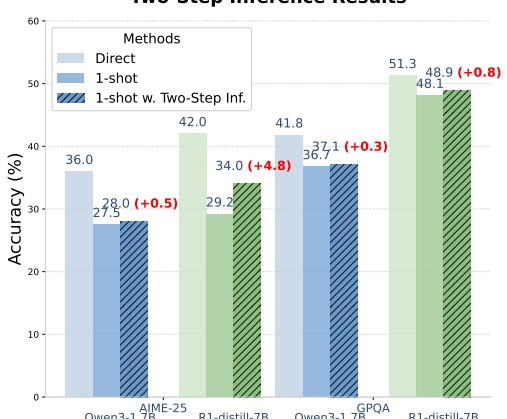

Figure 4: The results of two-step inference across on AIME'25 and GPQA. The blue color is for Qwen3-1.7B and the green color is for R1-Distill-Qwen-7B.

benchmarks, as shown in Figure 4. While it does not yet match the direct inference setting, the results are promising and drive us to study how to improve the quality of the reasoning trace in the the next section. We leave more case studies in Appendix B.4.

## 4 INSIGHT–REFINE–SOLVE

Section 3.3 shows that naive few-shot prompting can degrade RLM performance via two failure modes: semantic misleading, copying demonstration content by conflating superficially similar questions; and strategy transfer failure, failing to extract and apply the underlying problem-solving strategy. To turn demonstrations into assets rather than pitfalls, we introduce **I2S** (*Insight-to-Solve*), a test-time inference pipeline that extract reasoning examples into abstract guidance and then apply it to target questions. This method is naturally a two-step inference, breaking the shortcut between demonstrations and answers. We further introduce **I2S+**, an extension that integrates iterative self-refinement to strengthen the quality of reasoning traces on especially challenging problems.

### 4.1 DESIGN PRINCIPLES

**(1) Extract structural insights and apply to target.** Rather than conditioning directly on a demonstration's raw CoT, I2S focuses on extracting the high-level reasoning strategies embedded in the example and applying these strategies explicitly as guidance. By prioritizing abstract strategy over verbatim content, the model is encouraged to internalize transferable solution patterns instead of mimicking superficial lexical cues.

**(2) Complement, rather than override, internal reasoning.** Second, we complement the model's internal reasoning instead of overriding it. Modern RLMs already possess strong latent reasoning ability; demonstrations should act as hints rather than rigid templates that suppress this ability. I2S$^+$ therefore treats reasoning trace generated from demonstrations as sources, but allows the model to iteratively refine its own reasoning trace for the target question. Inspired by techniques such as self-refinement (Madaan et al., 2023), this approach supplements problem-solving with external insights without overwriting the model's inherent reasoning process.

### 4.2 STRUCTURED REASONING TRANSFER

Guided by these principles, the I2S pipeline mediates between the demonstration examples and a new target problem through a sequence of structured stages. Given a reasoning demonstration consisting of a question $x_{ex}$, a step-by-step reasoning trace $z_{ex}$, and its answer $y_{ex}$, along with a target question $x$, the model proceeds as follows:

**Comparison generation.** The model first produces a structured *comparison* $c \sim p(c \mid x, x_{ex})$ that highlights similarities and differences between the demonstration question $x_{ex}$ and target question $x$. For example, it might note: "both involve modular arithmetic, but the target introduces a parity constraint". This explicit contrast helps the model identify which aspects of the demonstration are relevant and where the problems diverge, reducing the risk of treating them as identical and falling into semantic lures. The full comparison-generation prompt is provided in Appendix C.4.

**Analysis derivation.** To obtain reusable insights for the target question, the model generates an *analysis* $a \sim p(a \mid c, z_{ex})$ conditioned on $c$ and $z_{ex}$. Irrelevant details are filtered out, while transferable strategies (e.g., modular reduction, enumeration) are retained. The analysis thus specifies *what* techniques should transfer, mitigating strategy extraction failure. The full analysis-derivation prompts are provided in Appendices C.5 and C.6.

**Reasoning generation.** Finally, the model produces its own reasoning trace $z \sim p(z \mid x, a)$, by applying analysis $a$ to the target question $x$. The demo reasoning trace $z_{ex}$ is withheld, forcing the model to construct its solution independently. In this way, we force the model to derive the answer through its own reasoning process, guided only by the abstracted insights, I2S directly counters the semantic misleading issue by preventing verbatim reuse of demonstration steps, and it addresses the strategy-transfer issue by explicitly injecting useful strategies into the reasoning process. The full reasoning-generation prompt is provided in Appendix C.3.

### 4.3 ITERATIVE SELF-REFINEMENT

For particularly difficult queries, a single pass of the above insight-to-solution pipeline may still produce an imperfect reasoning trace $z$ (for instance, it might contain minor errors or gaps in logic). The I2S+ variant augments the base pipeline with an iterative self-refinement loop that enables the model to gradually improve its reasoning. Each refinement iteration consists of three phases:

Figure 5: Three prompting paradigms. For *Two-step inference* (right), the demos are used only to elicit a trace; the demos are then removed and the final answer is generated from the question plus the elicited trace, decoupling answer generation from demo content.

**Suggestion.** Starting from the current reasoning trace $z$ for the target question, the model generates a set of candidate suggestions for modification or extension, such as alternative intermediate steps, sanity checks, or error corrections. They are produced in a structured format which can be systematically evaluated. Any invalid or trivial suggestions (e.g., contradictory or empty content) are discarded. The full suggestion prompt is provided in Appendix C.7.

**Review.** The model reviews the remaining suggestions by incorporating them back into the context and generating *check* statements that assess the quality or validity of each suggestion. Essentially, the model is asked to critique its own proposed changes. We may perform multiple rounds of review with varying randomness (controlled by temperature) to increase the chance of identifying a high-quality revision. The result of the review is an evaluation of which candidate suggestion is most promising for improving the reasoning. The full review prompt is provided in Appendix C.8.

**Refinement.** Finally, the model takes the best-rated suggestion (based on the checks) and uses it to revise the current reasoning trace $z$, yielding an updated $z'$. The revision is done in a controlled manner to improve coherence and correctness while preserving consistency with the question. If the edited trace fails to satisfy certain format or consistency constraints, the refinement step can be retried or adjusted. After this, the pipeline returns to the suggestion phase for another iteration, unless a stopping criterion is reached (e.g., a fixed number of iterations $N_{\text{iter}}$ or no further improvement). The full refinement prompt is provided in Appendix C.9.

By iteratively suggesting improvements, reviewing them, and refining the solution, I2S+ can detect and correct errors in the reasoning trace that might have eluded a single-pass solution. This self-correction loop further ensures that the final answer derivation is based on a sound and question-aligned reasoning process.

## 5 EXPERIMENTS

### 5.1 SETUP

**Datasets and models.** In order to have a comprehensive evalution, we extend the experiment setting based on the setting in Section 3.1. Except the close-ended benchmarks AIME 2025 (Mathematical Association of America, 2025) and GPQA Diamond (Rein et al., 2024), we conduct evaluation on open-ended General Reasoning task, which is constructed from the GeneralThought (GeneralReasoning, 2025) covering a broad spectrum of commonsense and domain-general reasoning tasks. We focus on two subsets: the Engineering domain, which contains technical and problem-solving oriented questions, and the General domain, which spans all categories of reasoning tasks. From these domains, we construct a balanced evaluation set of randomly selected 500 problems to measure open-ended reasoning ability. For models, We test three representative open-sorce RLMs spanning different scales: Qwen3-1.7B (Yang et al., 2025), DeepSeek-R1-Distill-Qwen-7B & 14B (Guo et al., 2025); and two closed-source models, GPT-4.1 and GPT o1-mini, to evaluate the effectiveness of the proposed method while keeping API costs manageable. For reasoning demonstrations dataset, we construct a demonstrations bank from OpenThoughts-114k (open-thoughts, 2024) and GeneralThought-430k (GeneralReasoning, 2025). To avoid leakage, the portion of GeneralThought used for the General Reasoning benchmark is strictly excluded from the retrieval database. We leave the implementation details in Appendix A.2. For all experiments, we run 5 times and report the average performance.

Table 1: Accuracy (%) on AIME25 and GPQA. Best per model & dataset in **bold**.

| Model | AIME25 | | | | | GPQA | | | | |
|---|---|---|---|---|---|---|---|---|---|---|
| | Direct | SR | PHP | I2S | I2S⁺ | Direct | SR | PHP | I2S | I2S⁺ |
| R1-Distill-Qwen-7B | 42.00 | 45.34 | 43.32 | 48.00 | **51.33** | 51.26 | 54.14 | 53.44 | 53.30 | **55.16** |
| R1-Distill-Qwen-14B | 50.00 | 56.67 | 58.00 | 54.00 | **60.00** | 59.50 | 60.71 | 60.40 | 59.60 | **63.03** |
| Qwen3-1.7B | 35.96 | 34.66 | 37.31 | 41.32 | **42.65** | 38.99 | 40.60 | 41.62 | 40.30 | **42.42** |
| Qwen3-14B | 72.00 | 73.33 | 74.67 | 73.32 | **76.00** | 62.52 | 62.73 | 62.63 | 63.54 | **66.06** |

Table 2: Accuracy (%) on General Reasoning. Best per model & dataset in **bold**.

| Model | General Reasoning - Eng | | | | | General Reasoning - General | | | | |
|---|---|---|---|---|---|---|---|---|---|---|
| | Direct | SR | PHP | I2S | I2S⁺ | Direct | SR | PHP | I2S | I2S⁺ |
| R1-Distill-Qwen-7B | 40.04 | 40.84 | 40.72 | 40.24 | **41.36** | 51.28 | 50.52 | 51.36 | **51.72** | 49.28 |
| R1-Distill-Qwen-14B | 49.44 | 49.88 | 50.08 | 51.04 | **53.28** | 57.68 | 61.16 | 61.00 | 66.64 | **68.40** |
| Qwen3-1.7B | 43.04 | 39.88 | 41.08 | 43.40 | **43.44** | 55.08 | 51.52 | 52.76 | **57.16** | 56.72 |
| Qwen3-14B | 66.88 | 67.20 | 68.08 | 72.04 | **72.20** | 78.56 | 79.88 | 80.00 | 84.92 | **85.12** |

**Methods to compare.** We compare three inference strategies against our method: Direct Inference, a zero-shot baseline without demonstrations; Self-Refine (SR) (Madaan et al., 2023), which iteratively critiques its own draft and revises the answer by re-feeding the problem, initial solution, and self-generated feedback for a few rounds; and Progressive-Hint Prompting (PHP) (Zheng et al., 2023a), which supplies increasingly specific hints only as needed until the problem is solved or all hints are exhausted. We leave more details about baseline methods in Appendix A.1. Our evaluation pipeline is mainly implemented based on LightEval (Habib et al., 2023). We leave more implementation details and decoding hyperparameters in Appendix A.2.

## 5.2 MAIN RESULTS

Tables 1 and 2 summarize performance on both *closed-ended* (AIME'25, GPQA) and *open-ended* general reasoning. Overall, our methods, I2S and I2S⁺, consistently outperform Direct inference and the sequential test-time scaling baselines (SR, PHP). For example, on AIME'25, I2S⁺ improves R1-Distill-Qwen-14B from 50.00 to 60.00 (**+10.0**), and on GPQA it raises the same model from 59.50 to **63.03** (**+3.5** ). These gains over both Direct and SR/PHP indicate that explicitly decoupling *insight extraction* from *solution drafting* and then self-refining the draft is particularly effective when answers are verifiable. Further, we provide case studies about with/without the proposed method in Appendix B.4.3. On general reasoning, both I2S and I2S⁺ outperform Direct and are better than SR/PHP; the gains are smaller. And I2S at times matches or even slightly outperforms I2S⁺; we hypothesize this is partly due to the sensitivity of LLM-as-judge effects (e.g., style and brevity).

## 5.3 HOW EFFICIENTLY DOES EXTRA COMPUTE TURN INTO GAIN?

We aim to study *how effectively extra test-time compute turns into accuracy*. majority@$N$ (Wang et al., 2022) is a parallel test-time scaling baseline: sample $N$ independent solutions, extract and canonicalize the final answers, then pick the most frequent. While majority@$N$ is ill-defined for *open-ended* generation and not strictly comparable to our sequential scaling scheme, on *closed-ended* tasks, it has demonstrated strong *return on compute*: additional parallel decoding translates efficiently into accuracy gains at small $N$.

Rather than matching raw FLOPs, we count *question-conditioned calls*—forward passes whose inputs include the target question. In I2S, only three stages directly condition on the question (comparison, analysis, and final answer generation), so the model "sees" the question three times. We therefore compare I2S to majority@3, which uses the same number of question-conditioned calls but in parallel. As Table 3 shows, I2S consistently outperforms majority@3 on AIME'25 and GPQA, indicating better compute-to-gain efficiency at small budgets. We also compare I2S+ to a stronger parallel baseline, majority@32, which uses a roughly comparable number of question-conditioned calls. On AIME'25, majority@32 is slightly higher across models; on GPQA, I2S+ surpasses majority@32 for the R1-Distill-Qwen 14B and Qwen3-1.7B models and is close for R1-Distill-Qwen 7B. Overall, our methods I2S and I2S+ converts test-time compute into utility at least as efficiently as brute-force sampling and voting, and often more so at modest budgets.

Table 3: Comparison between I2S and I2S+ and majority@3 and 32 across close-ended benchmarks.

| Model | AIME25 | | | | GPQA | | | |
|-------|--------|-----|--------|------|--------|-----|--------|------|
| | Maj@3 | I2S | Maj@32 | I2S+ | Maj@3 | I2S | Maj@32 | I2S+ |
| R1-Distill-Qwen-7B | 46.66 | **48.00** | **52.00** | 51.33 | 53.13 | **53.30** | **55.66** | 55.16 |
| R1-Distill-Qwen-14B | 51.33 | **54.00** | **62.00** | 60.00 | 59.50 | **59.60** | 60.31 | **63.03** |
| Qwen3-1.7B | 39.33 | **41.32** | **44.66** | 42.65 | 40.10 | **40.30** | 41.92 | **42.42** |
| Qwen3-14B | 73.30 | **73.32** | 75.00 | **76.00** | 62.80 | **63.54** | 63.62 | **66.06** |

Table 4: Performance across different Refinement Iterations. Results are averaged over 5 runs (means shown); bold marks the best iteration per row.

| Dataset | Model | I2S (Iter 0) | Iter 1 | Iter 2 | Iter 3 |
|---------|-------|--------------|--------|--------|--------|
| AIME25 | R1-Distill-Qwen-7B | 48.00 | 50.00 | **51.33**(+3.33) | 51.33 |
| | R1-Distill-Qwen-14B | 54.00 | 59.33 | 59.33 | **60.00**(+6.00) |
| | Qwen3-1.7B | 41.32 | **42.65**(+1.33) | 40.65 | 41.99 |
| | Qwen3-14B | 73.32 | 73.33 | 74.67 | **76.00**(+2.68) |
| GPQA | R1-Distill-Qwen-7B | 53.30 | **55.16**(+1.86) | 54.66 | 54.05 |
| | R1-Distill-Qwen-14B | 59.60 | 60.40 | 61.21 | **63.03**(+3.43) |
| | Qwen3-1.7B | 40.30 | 41.21 | 41.61 | **42.42**(+2.12) |
| | Qwen3-14B | 63.54 | 65.35 | **66.06**(+2.52) | 65.56 |
| Engineering | R1-Distill-Qwen-7B | 40.04 | **41.36**(+1.32) | 40.40 | 39.16 |
| | R1-Distill-Qwen-14B | 51.04 | 51.28 | 53.28 | **53.28**(+2.24) |
| | Qwen3-1.7B | 43.04 | 43.12 | **43.44**(+0.40) | 42.68 |
| | Qwen3-14B | 72.04 | 71.04 | **72.20**(+0.16) | 71.36 |
| General | R1-Distill-Qwen-7B | 51.72 | 48.88 | **49.28**(-2.44) | 48.20 |
| | R1-Distill-Qwen-14B | 66.64 | 67.80 | 67.84 | **68.40**(+1.76) |
| | Qwen3-1.7B | 57.16 | 56.00 | **56.72**(-0.44) | 56.64 |
| | Qwen3-14B | 84.92 | **85.12**(+0.20) | 84.28 | 83.96 |

## 5.4 EFFECT OF REFINEMENT ITERATIONS

In order to study the effect of iterarive refinement process within the I2S+, We evaluate I2S+ with up to three refinement iterations and record performance after each step (Iter 0 denotes I2S without refinement). As shown in Table 4, I2S+ with refinement delivers clear gains on math—showing strong scale effects, e.g., on AIME'25, refinement shows clear early returns: most of the gains appear in the first 1–2 iterations and then saturate (7B: +3.33; 14B: +6.00; 1.7B peaks at Iter 1 and regresses thereafter). On GPQA, behaviors are model-dependent: 7B peaks at Iter 1 (+1.86) and then softens, whereas 14B improves monotonically to +3.43 by Iter 3; 1.7B rises gradually. For open-ended Engineering/General under LLM-as-judge, changes are marginal or occasionally negative, likely because single-reference grading and weak external feedback provide noisy or sparse signals for refinement rather than indicating intrinsic limits of the method. We leave better judges (multi-reference/tolerant scoring) and explicit guidance signals for future work; detailed discussions are in Appendix B.2.

## 5.5 COMPUTATION-PERFORMANCE TRADE-OFF

A central question in test-time scaling is: *how much additional inference-time compute do we need to pay for a given performance gain?* To study this trade-off, we measure, for each dataset and model, the average total number of *generated* tokens per question over five runs. For each run, we use the trajectory that achieves the accuracy reported in the previous tables and record its total number of generated tokens. Figure 6 then plots accuracy as a function of the generated-token budget (our proxy for computation cost). We use the number of generated tokens as a proxy for computation cost because, in vLLM, the prefill phase (processing the input prompt) is highly optimized and relatively fast, while most of the wall-clock time is spent in autoregressive decoding. Consequently, the total number of generated tokens closely tracks both latency and FLOPs at test time. Across two models (R1-Distill-Qwen-7B and Qwen3-14B) and two benchmarks (AIME'25 and GPQA), our method achieves a more favorable computation–performance trade-off than the baselines: for a fixed token budget, it consistently matches or outperforms competing methods in accuracy; conversely, to reach a given accuracy level, it requires substantially fewer generated tokens.

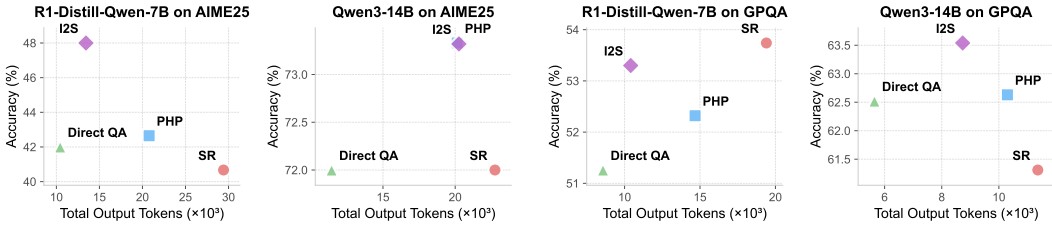

Figure 6: Computation cost (number of generated tokens) vs. performance on AIME'25 and GPQA.

## 5.6 FAILURE MODE STUDY

To understand *how* I2S mitigates the failures identified in Section 3.3, we conduct a failure-mode analysis on AIME'25 and GPQA. Concretely, we use GPT-5.1 to inspect the reasoning traces same as in Section 5.5 and annotate each incorrect case into one of three categories: **(i) semantic misleading**, where the model copies or overfits to the demonstration and treats the target as essentially the same problem; **(ii) transfer failure**, where it fails to extract and apply the underlying reasoning strategy; and **(iii) other errors**, including calculation mistakes and genuinely incorrect internal reasoning (i.e., hard questions that are simply beyond the capability of this model). Figure 7 compares one-shot in-context prompting with I2S across two models (R1-Distill-Qwen-7B and Qwen3-1.7B). Across all four settings, I2S substantially reduces demonstration-induced errors: the share of *semantic misleading* drops by roughly 15 to 33 points, and *transfer failure* also decreases. The remaining errors shift into the "other" category, indicating that I2S largely **neutralizes the harmful influence of demonstrations** and leaves a residual set of more standard, heterogeneous reasoning mistakes instead. This aligns with our case studies and supports the view that I2S directly targets the two core failure modes of in-context reasoning.

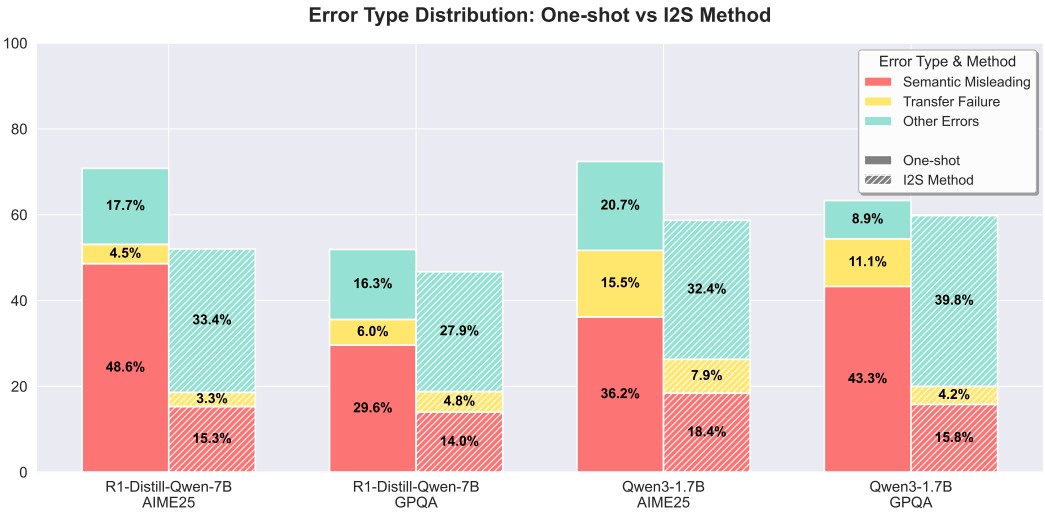

Figure 7: Failure-mode breakdown of one-shot in-context reasoning vs. I2S.

## 6 CONCLUSION

In this work, we revisited the observation that modern RLMs can perform worse with few-shot reasoning demonstrations than with direct answering, even when demonstrations are high quality and closely matched to the target question. Our analysis attributes this degradation to two mechanisms: *semantic misleading*, where surface similarity induces near-verbatim copying, and *strategy transfer failure*, where useful problem-solving patterns are not extracted and applied to the target. To convert demonstrations from poison into assets, we proposed Insight–Refine–Solve framework, which extracts transferable insights from demonstrations (I2S), optionally self-refines the target-specific reasoning trace (I2S+), and generated solution in decoupled way. Across closed-ended benchmarks (AIME'25, GPQA) and open-ended General Reasoning, I2S/I2S+ consistently outperform direct answering and baselines across diverse models, with gains extending even to closed-source GPT series (e.g., GPT-4.1, o1-mini). Our work suggests that reasoning demonstrations are not inherently detrimental to RLMs; when used properly, they improve rather than degrade performance.

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

## A  EXPERIMENT SETTING

### A.1  METHODS TO COMPARE

We compare the following inference methods, including our proposed approach:

**Direct Inference.** Standard zero-shot direct answering without demonstrations. This serves as the simplest baseline to measure the model's inherent reasoning ability. The prompt can be found in C.1.

**Self-Refine (SR)** (Madaan et al., 2023) This method improves initial outputs through a process of iterative feedback and refinement. The model first generates an initial solution to a problem. Then, in the feedback step, the same model is prompted to provide a critique of its own solution, identifying any errors or areas for improvement. Finally, in the refinement step, the original problem, the initial solution, and the generated feedback are all concatenated and provided as a new input to the model, which then generates a refined answer. This feedback-refinement loop can be repeated multiple times to further enhance the quality of the solution.

We implement the standard feedback-refinement loop and evaluate up to $4$ rounds per example, reporting the highest per-round score under the task metric. The maximum iteration count of $4$ in SR aligns with Madaan et al. (2023). To control context length and avoid exceeding token constraints, iteration prompts exclude model-internal traces (content delimited by <think>...</think>); only the answer part from the previous round is carried forward. Each round uses two prompts: a feedback prompt, "Is this answer reasonable and correct? Please provide feedback for the Round {index} Answer.", followed by a refinement prompt, "Please provide your refined answer based on the above content".

**Progressive-Hint Prompting (PHP)** (Zheng et al., 2023a) This method utilizes a series of hints to guide the model toward the correct answer. The process is as follows: The model is first given the problem and a high-level hint. If the model fails to produce the correct answer, it is provided with the same problem but with a more specific hint. This continues until the model either solves the problem or all hints have been exhausted. The key idea is to provide just enough information to correct the model's reasoning without giving away the final answer.

We adopt a progressive-hinting procedure, where the model is re-invoked with the same question plus a compact trail of its previous externally visible attempts as guidance; we cap the process at 3 retries and report the best per-round score under the task metric. To keep the prompts compact, the hint trail omits any content <think>...</think>. Each retry begins with "Here are your previous attempts:", enumerates entries such as "Round {i}: {previous_answer}", and ends with "Please try again." This setup preserves the core idea of progressively steering the model using its own observable outputs while controlling prompt length and ensuring a uniform fixed-budget selection protocol.

**Majority Voting (Maj@$N$)** (Wang et al., 2022) Unlike our sequential test-time scaling approach, Maj@$N$ performs *parallel* scaling and is therefore not strictly comparable. It is naturally suited to closed-ended tasks (e.g., multiple choice); extending it to open-ended generation typically requires an additional aggregation step—prompting the model (or a judge) to read the $N$ candidates and synthesize a single answer. *Out of curiosity*, we examine whether, under matched test-time compute, our sequential scaling offers better utility than parallel Maj@$N$. We acknowledge that this comparison may disadvantage our method on closed-ended tasks, but we include it to contextualize the trade-offs.

### A.2  IMPLEMENTATION DETAILS

For retrieval-augmented generation, we construct a FAISS index of query–reasoning–answer triples with all-mpnet-base-v2 embeddings, and include the nearest exemplar's reasoning trace as additional context to the generator.

All inferences are performed using the vLLM engine with default sampling parameters unless otherwise noted (temperature = 0.5, top-p = 0.95, maximum context length up to 32k tokens). In rare cases where the model hallucinates or fails to follow the required output format, we fall back to a simple callback that directly requests the final answer.

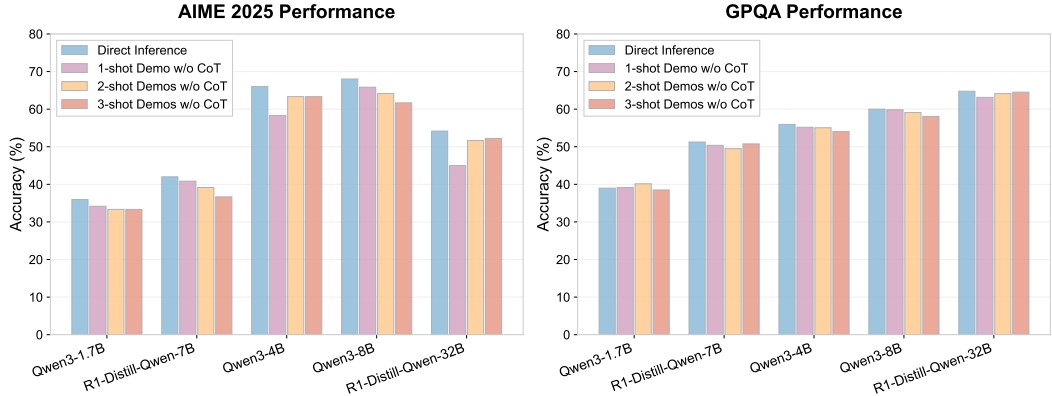

Figure 8: In-context learning with demonstrations that omit explicit thinking steps also harms the performance of reasoning language models (RLMs).

For evaluation, we adopt two complementary strategies, depending on the dataset. For AIME25 and GPQA, we use the LightEval toolkit (Habib et al., 2023) for standardized assessment.

For Engineering and General reasoning tasks, we employ a two-step LLM-judge pipeline similar to (Zheng et al., 2023b). Specifically:

1. **Reference answer generation.** Each question is first solved by GPT-4.1 to produce a gold reference answer.

2. **Answer extraction.** Since model outputs often contain full reasoning traces or invisible tags (e.g., `<think>`), they are processed with an extraction prompt that instructs GPT-4.1 to return only the canonical answer in the required format.

3. **Answer verification.** The extracted student answer is then compared against the reference using GPT-4.1 as the judge. The judging prompt enforces strict equivalence checking and produces a binary decision ("Final Decision: True/False").

The corresponding LLM-as-Judgment prompts can be found in Appendix C.1 and C.2.

## B  EXPERIMENT RESULTS AND ANALYSIS

### B.1  REASONING MODELS IN-CONTEXT LEARNING WITH NON-THINKING DEMOS

In the Section 1, we present a puzzling observation: the very training samples used to distill R1-Distill-Qwen-7B, when reused as few-shot reasoning demonstrations, can noticeably decrease its performance. This behavior is at odds with the common view of in-context learning as a gradient-descent-like mechanism on those same examples (Von Oswald et al., 2023; Ahn et al., 2023; Dai et al., 2022).

A further question is whether this degradation is specific to demonstrations containing explicit chain-of-thought, or whether it also arises when the model is conditioned on *non-thinking* demonstrations (i.e., examples that provide the final answer without intermediate reasoning steps). In this section, we therefore study the in-context learning behavior of reasoning models under non-thinking demonstrations. As shown in Figure 8, we observe that, similar to the case with thinking demos, the performance of the reasoning model drops when non-thinking demos are used. Moreover, as we increase the number of demonstrations, the performance continues to worsen and never recovers to the level of direct inference (without demonstrations).

### B.2  DISCUSSION FOR ITERATIVE REFINEMENT

Table 4 summarizes the effect of adding iterative refinement (I2S+) on top of our base method (I2S). In general, we find clear benefits on mathematical reasoning, early saturation of gains with additional iterations, and mixed outcomes under LLM-as-judge evaluation for open-ended tasks.

**Iterations boost mathematical reasoning, with clear scale effects.** On AIME 2025, refinement consistently improves over I2S, with larger models reaping greater benefits. Specifically, *R1-Distill-*

*Qwen-7B* increases from 48.00 to **51.33**, *R1-Distill-Qwen-14B* from 54.00 to **60.00**, and *Qwen3-1.7B* from 41.32 to **42.65**. These correspond to gains of +3.33, +6.00, and +1.33 points, respectively, suggesting that larger models can exploit additional reasoning passes more effectively (e.g., by correcting intermediate mistakes or refining derivations).

**Most gains arrive early, with diminishing returns thereafter.** The first refinement iteration delivers the bulk of the improvement, while subsequent iterations yield marginal returns. On AIME 2025, the 7B model obtains roughly two-thirds of its total gain after a single iteration, and the 14B model achieves nearly all of its +6.00 improvement in the first pass. Beyond 1–2 refinement rounds, performance typically *plateaus by iteration 2 or 3*: the best scores for *R1-Distill-Qwen-7B* and *R1-Distill-Qwen-14B* occur at iteration 3 (with only tiny increments over iteration 2), while the smallest model (*Qwen3-1.7B*) peaks at iteration 1. This indicates that one or two cycles capture most of the available corrective signal.

**GPQA: modest gains, and caution against over-iteration.** On GPQA, iterative refinement yields smaller improvements and the optimal iteration count depends on model size. R1-Distill-Qwen-7B rises from 53.30 to 55.16 at iteration 1 (+1.86), but further iterations do not help (54.66 at iteration 2) and even soften to 54.05 by iteration 3. In contrast, R1-Distill-Qwen-14B improves monotonically from 59.60 to 63.03 (+3.43), while Qwen3-1.7B progresses from 40.30 to 42.42 by iteration 3 (41.21 at iteration 1, 41.61 at iteration 2). In practice, one refinement often suffices for medium-sized models on factual QA, while very small models can benefit from a few extra passes—albeit with modest returns.

**Under LLM-judged tasks, refinement yields reliable gains only with sufficient scale.** With a GPT-4.1 judge, R1-Distill-Qwen-14B consistently benefits from additional passes, while smaller models are flat or regress. On Engineering, 14B moves from 51.04 to 53.28 (+2.24), plateauing by iterations 2–3; 7B nudges up at iteration 1 (40.04 $\to$ 41.36, +1.32) then declines to 39.16 by iteration 3; 1.7B shows only a transient +0.40 at iteration 2. On General, 14B climbs steadily (66.64 $\to$ 68.40, +1.76), whereas 7B degrades overall (51.72 $\to$ 48.20) and 1.7B hovers near 56–57% without a durable trend. When supervision is implicit (agreement rather than verifiable correctness), extra passes tend to amplify noise unless the model is large enough to preserve substance while tightening form; at 14B, refinement is therefore useful and reliable, but at 7B/1.7B it is fragile and often counterproductive.

**Why do some tasks show weak or negative refinement gains?** *Multiple factors are at play.* (1) *Open-endedness and reference mismatch:* Engineering/General prompts often admit many valid realizations. Evaluation against a single reference answer (authored by GPT-4.1) can penalize refined responses that are correct but differ in phrasing or approach. (2) *Lack of a clear corrective signal:* Iterative refinement is most effective when objective errors can be identified and fixed (as in math). In agreement-based grading, the feedback is implicit and noisy, so extra passes may induce superficial edits or stylistic drift rather than substantive gains. (3) *Evaluation side-effects:* To mitigate length bias, we apply an extraction step that keeps only "key parts" before scoring. For prompts expecting visible reasoning, this truncation can remove context the judge values, inadvertently turning improved answers into false negatives. (4) *Scale-dependent utilization of weak signals*: Larger models more reliably infer task intent and convert weak/implicit feedback into minimal, goal-aligned edits: they retain high-salience content, keep intermediate checks consistent, and shape responses to the question's stated requirements without distorting substance. This capacity underpins the monotonic 14B gains on GPQA/Engineering/General. By contrast, 7B/1.7B show unstable intent adherence across passes—edits drift from the objective, checks are applied inconsistently, and crucial material is dropped or diluted—so additional passes compound small deviations rather than correct them, yielding flat or negative returns.

**Toward better refinement in open-ended tasks.** These observations do not imply that iterative reasoning is ineffective for open-ended problems; rather, they underscore the need for more nuanced strategies. Future work includes (i) using multiple references or more tolerant scoring criteria to fairly assess semantically equivalent refinements, (ii) incorporating explicit feedback (e.g., judge-guided hints) to provide a stronger corrective signal, and (iii) refining extraction/scoring so essential reasoning is preserved.

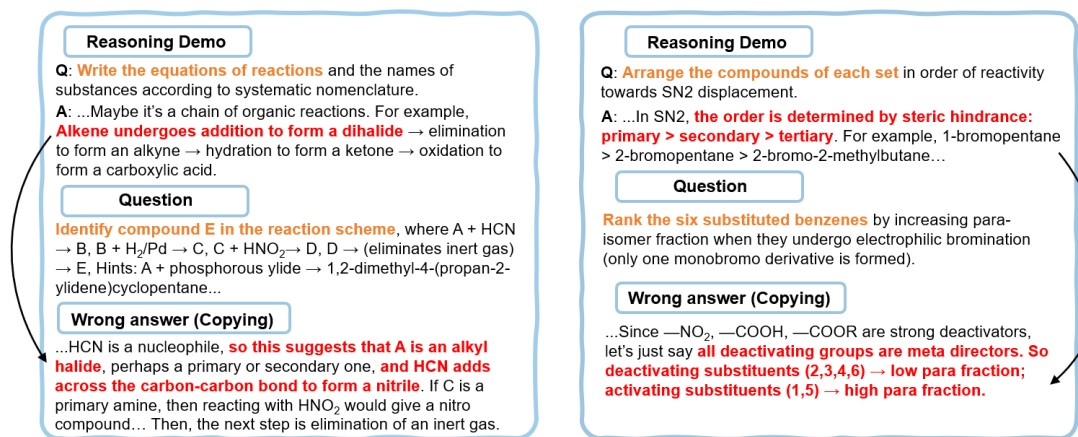

Figure 9: Failure modes in failure to transfer underlying reasoning structure. (red: inappropriate copying).

## B.3 CONTEXT LENGTH ANALYSIS

We measure the effective context length across the pipeline. While intermediate stages (such as comparison and analysis) can flexibly adjust their output length, the reasoning demonstration (CoT trace) remains the dominant contributor to overall context size. Table 5 reports average and maximum token lengths for exemplar components in both the AIME25 and GPQA datasets. Notably, the reasoning demonstrations are substantially longer than the other components, with maximum lengths exceeding 13k tokens in AIME25 and 14k tokens in GPQA. Such excessively long contexts risk introducing confusion and degrading reasoning quality. This observation motivates our design choice of restricting to a single exemplar per query, in order to stabilize generation and control context growth.

Table 5: Average and maximum token length of exemplars for AIME25 and GPQA datasets

| Key | AIME25 | | GPQA | |
|---|---|---|---|---|
| | Avg | Max | Avg | Max |
| demonstration question | 111.63 | 241 | 101.49 | 426 |
| reasoning demonstration (CoT trace) | 6141.83 | 13813 | 2994.32 | 14969 |
| demonstration answer | 751.20 | 1758 | 152.03 | 2097 |
| target question | 154.47 | 750 | 208.75 | 2763 |

## B.4 MORE CASE STUDY

In addition to the main cases discussed in Section 3.3, we further extend our analysis with examples from chemistry and physics, which reinforce the generality of two characteristic failure modes of in-context reasoning: *Semantic Misleading from Demonstration Content* and *Failure to Transfer Underlying Reasoning Structure*. These additional cases highlight that the same vulnerabilities persist across domains, emphasizing the model's tendency to overfit to surface cues and its difficulty in transferring deeper reasoning schemas.

### B.4.1 SEMANTIC MISLEADING FROM DEMONSTRATION CONTENT

In Figure 9, the model was misled by superficial linguistic overlap with the demonstration. In the first case, the demonstration presented a canonical chain reaction sequence "alkene → dihalide → alkyne → ketone → carboxylic acid," which primed the model to associate any structure of the form "A → B → C → D → E" with this heuristic. Faced with a target problem requiring compound identification from spectroscopy and elimination steps, the model incorrectly inferred that "HCN is a nucleophile, so A must be an alkyl halide, yielding a nitrile," ignoring the IR carbonyl signature and

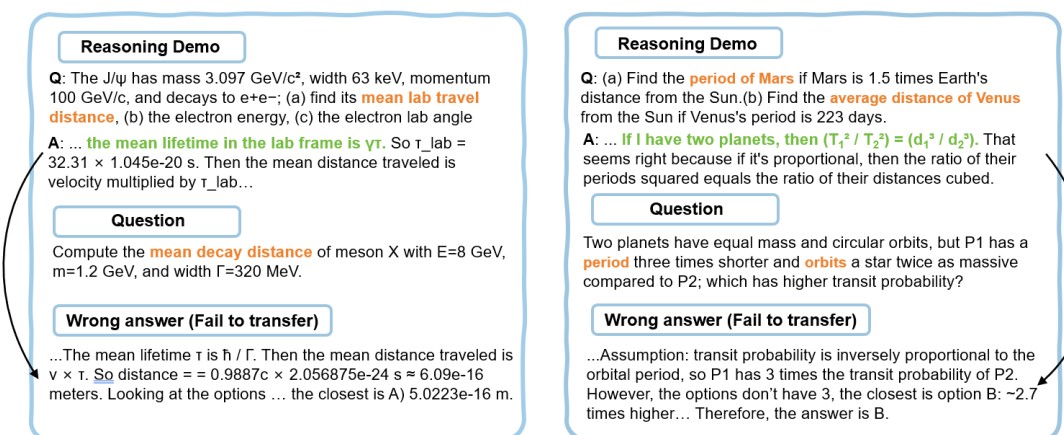

Figure 10: Failure modes in semantic misleading from demonstration content. (green: missed transferable insight).

diazotization constraints. Similarly, in the second case, the shared phrase "arrange the compounds in order" triggered the transfer of the heuristic "SN2 reactivity is ordered by steric hindrance: primary > secondary > tertiary." The model directly transplanted this rule into a substituted benzene problem, asserting that "all deactivating groups are meta directors" and partitioning substituents into a simplistic binary classification, while failing to account for exceptions such as halogens. In both cases, the demonstrations acted as semantic lures, where surface lexical overlap obscured the structural requirements of the target tasks.

### B.4.2 Failure to Transfer Underlying Reasoning Structure.

By contrast, in Figure 10 the demonstrations contained genuinely useful structural insights, yet the model failed to transfer them correctly. In the particle decay example, the demonstration clearly outlined the three-step chain: compute the rest-frame lifetime $\tau_{\text{rest}} = \hbar/\Gamma$, apply time dilation with $\tau_{\text{lab}} = \gamma\tau_{\text{rest}}$ using $\gamma = E/m$, and then obtain the decay distance $d = \beta c\tau_{\text{lab}}$. The target required exactly this reasoning, but the model collapsed the structure, treating $\tau_{\text{rest}}$ as if it were $\tau_{\text{lab}}$ and computing $d = \beta c\tau_{\text{rest}}$, thereby underestimating the distance by a factor of $\gamma$. In the transit probability problem, the demonstration exploited Kepler's law ($T^2 \propto a^3$) to compare orbital scales. The target required chaining this with the geometric relation $P_{\text{tr}} \sim R_\star/a$ and substituting $a \propto M_\star^{1/3}P^{2/3}$. Instead, the model extracted only the middle fragment, simplifying the result to $P_{\text{tr}} \propto 1/P$ and concluding that the probability is three times higher rather than the correct factor $(9/2)^{1/3} \approx 1.65$. These cases show that while the demonstrations provide structurally sound reasoning patterns, the model often transfers only fragments, skipping critical steps and thereby producing systematically flawed conclusions.

### B.4.3 From Error to Correctness: A Case Study

To study how the proposed method turns failure into correctness, run with-vs-without I2S and present the following two cases::

As shown in Figure 11, the analysis step enhances reasoning by abstracting domain-specific operations into generalizable procedural guidance. Instead of incorrectly equating the squared inner product itself with probability, the analysis explicitly delineates the steps of normalization, eigenvector identification, projection, and squaring. This abstraction eliminates local computational shortcuts, enforces alignment with the formal definition of measurement probability, and ultimately yields the correct outcome of $1/3$. The case demonstrates that structured analysis not only corrects errors but also provides a principled framework that stabilizes inference across problem instances.

The same I2S template applies to another case in Figure 12. The case demonstrates how structured analysis corrects misinterpretations and guides toward the consistent formula. The wrong path arose from misreading the $1650 \text{ cm}^{-1}$ signal as a second carbonyl rather than a C=C stretch, and from

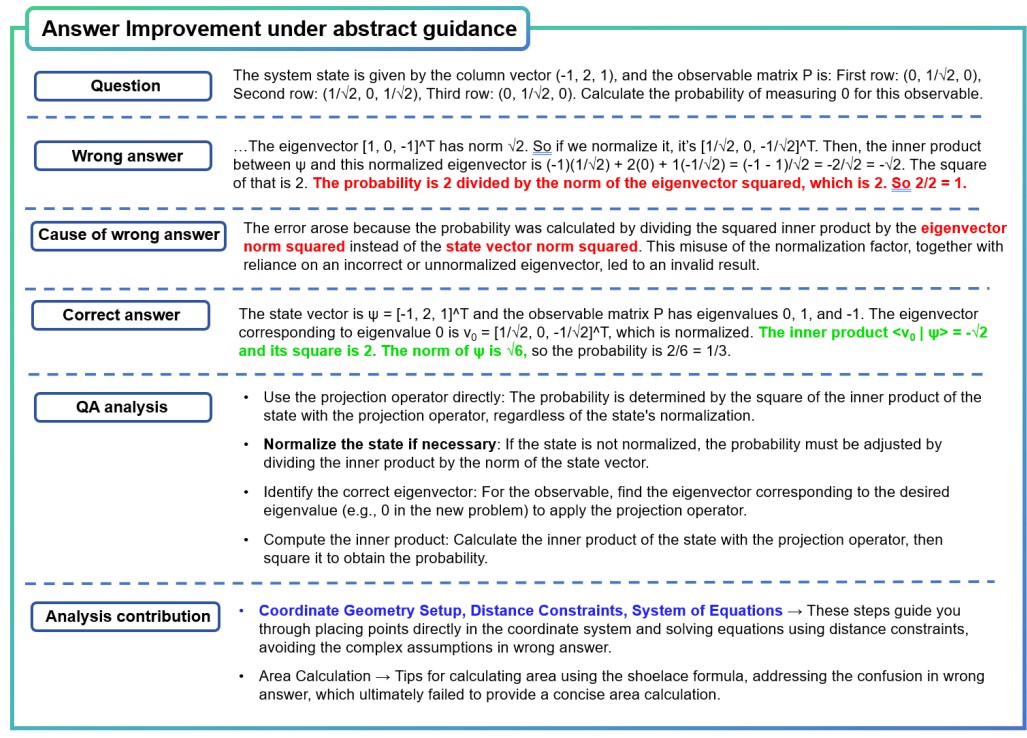

Figure 11: Abstract guidance corrects the answer. (red: wrong answer and the cause. green: correct answer. blue: key principles.)

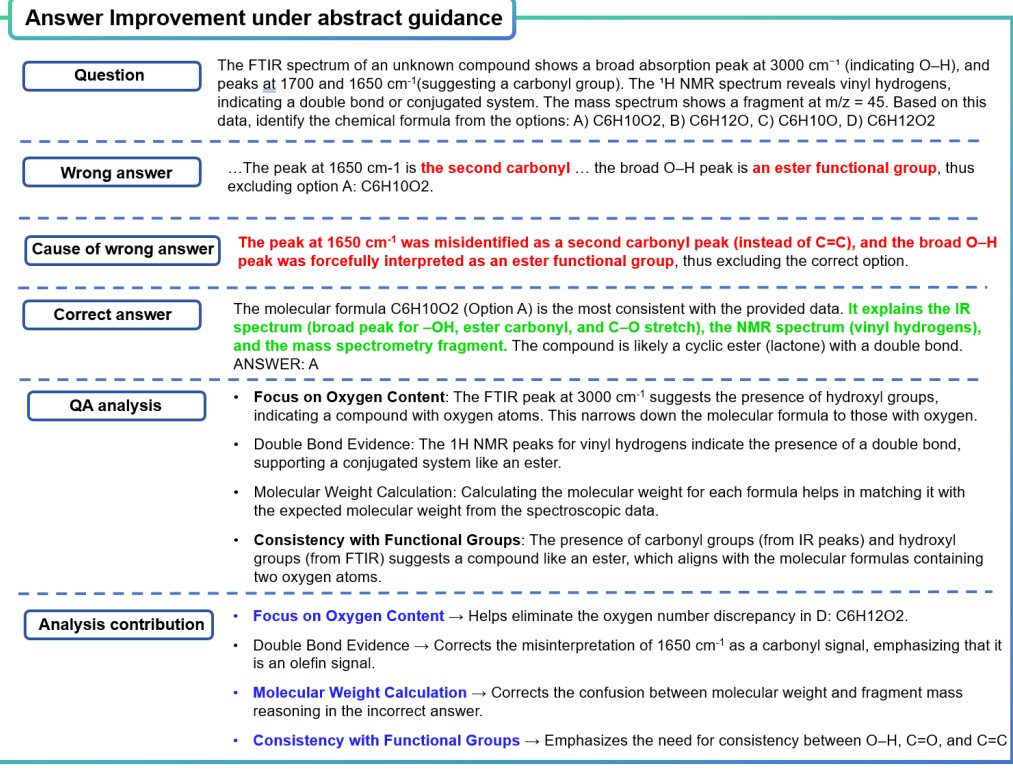

Figure 12: Abstract guidance corrects the answer. (red: wrong answer and the cause. green: correct answer. blue: key principles.)

forcing the O–H band into an ester interpretation. The refined analysis instead emphasizes oxygen balance, recognition of vinyl hydrogens as evidence of unsaturation, and consistency across IR, NMR, and MS. By integrating these checks, the method converges on $C_6H_{10}O_2$, a cyclic ester (lactone) with a double bond, fully reconciling the spectroscopic evidence. This illustrates how abstract guidance—oxygen count, bond-type assignment, molecular-weight reasoning, and functional-group consistency—stabilizes inference and prevents error propagation.

## B.5 STANDARD DEVIATION TABLE

The results reported in the main experiments represent the average performance across 5 runs. Table 6 through Table 9 present the corresponding standard deviations.

| Model | AIME25 | | | | | | |
| --- | --- | --- | --- | --- | --- | --- | --- |
| | Direct | SR | PHP | Maj@32 | I2S | I2S+ | Maj@3 |
| R1-Distill-Qwen-7B | 5.82 | 7.67 | 4.08 | 1.82 | 5.10 | 2.98 | 4.22 |
| Qwen3-1.7B | 2.50 | 2.99 | 4.35 | 2.98 | 1.81 | 2.79 | 3.88 |
| R1-Distill-Qwen-14B | 6.33 | 4.08 | 5.06 | 2.70 | 3.27 | 3.33 | 4.52 |
| Qwen3-14B | 5.81 | 2.36 | 3.80 | 2.46 | 4.21 | 1.49 | 4.18 |

Table 6: Standard deviation on AIME25.

| Model | GPQA | | | | | | |
| --- | --- | --- | --- | --- | --- | --- | --- |
| | Direct | SR | PHP | Maj@32 | I2S | I2S+ | Maj@3 |
| R1-Distill-Qwen-7B | 2.16 | 1.65 | 3.45 | 2.15 | 1.30 | 1.10 | 3.41 |
| Qwen3-1.7B | 2.05 | 2.54 | 2.62 | 2.17 | 1.49 | 3.31 | 1.22 |
| R1-Distill-Qwen-14B | 1.61 | 2.15 | 1.73 | 1.85 | 1.06 | 1.57 | 1.26 |
| Qwen3-14B | 1.37 | 1.45 | 1.42 | 2.18 | 2.60 | 0.90 | 1.33 |

Table 7: Standard deviation on GPQA.

| Model | General Reasoning (Eng) | | | | | | |
| --- | --- | --- | --- | --- | --- | --- | --- |
| | Direct | SR | PHP | Maj@8 | I2S | I2S+ | Maj@3 |
| R1-Distill-Qwen-7B | 1.28 | 1.87 | 1.08 | 1.51 | 2.51 | 0.90 | 1.03 |
| Qwen3-1.7B | 0.97 | 1.71 | 1.04 | 1.15 | 1.34 | 1.19 | 1.05 |
| R1-Distill-Qwen-14B | 0.54 | 1.86 | 1.49 | 2.56 | 0.96 | 0.59 | 0.97 |
| Qwen3-14B | 1.10 | 1.13 | 1.50 | 1.07 | 1.33 | 0.99 | 1.07 |

Table 8: Standard deviation on General Reasoning (Eng).

| Model | General Reasoning (General) | | | | | | |
| --- | --- | --- | --- | --- | --- | --- | --- |
| | Direct | SR | PHP | Maj@8 | I2S | I2S+ | Maj@3 |
| R1-Distill-Qwen-7B | 0.95 | 2.13 | 1.81 | 1.49 | 1.16 | 2.52 | 0.94 |
| Qwen3-1.7B | 0.65 | 1.51 | 1.44 | 0.81 | 1.23 | 0.94 | 0.78 |
| R1-Distill-Qwen-14B | 1.09 | 2.84 | 1.81 | 0.55 | 1.68 | 1.47 | 0.75 |
| Qwen3-14B | 1.28 | 1.43 | 0.62 | 0.55 | 0.86 | 1.10 | 1.05 |

Table 9: Standard deviation on General Reasoning (General).

## C PROMPT DESIGN

### C.1 GENERATION PROMPT

This prompt produces an answer candidate for a given question.

---

**Generation Prompt**

```
You are a knowledgeable problem solver.
Answer the following question and provide only the final answer in
    the format:
Final Answer: <your final answer>

Question:
{question}
```

---

### C.2 VERIFICATION PROMPT

This prompt verifies the generated answer against the gold label and returns a clean Boolean signal.

---

**Verification Prompt**

```
You are a logical and fair evaluator.

Your task:
Step 1: Compare the provided Final Answer with the Gold Answer.
Step 2: If the provided Final Answer is correct, directly output:
Answer: True
Otherwise, directly output:
Answer: False

---
Question:
{question}

Gold Answer:
{model_answer}

Final Answer:
{generated_answer}
```

---

### C.3 ANSWER CONSTRUCTION PROMPT

This prompt guides the model to perform step-by-step reasoning for a new question, leveraging example and comparison.

**Answer Construction Prompt**

```
You are an expert on general reasoning tasks.
Solve the new question step by step, using clear and logical
    reasoning.
Conclude with: 'Answer: $LETTER' (without quotes) where LETTER is
    one of ABCD.

You are given:
1. An example question
2. A comparison that identifies common reasoning patterns
3. Several potentially helpful insights
4. A new question that you are currently solving

Example Question:
{example_question}
{comparison}
{analyze}
New Question:
{question}

You may use the example and comparison to guide your reasoning if
    helpful.
Use only the insights that are relevant to the new question.
```

## C.4 COMPARISON PROMPT

This prompt encourages structural analysis between the example and the new question.

**Comparison Prompt**

```
You are provided with two questions:
Example Question:
{example_question}
New Question:
{current_question}

Your task is to analyze and compare the two questions. Focus on:
1. Structural similarities or differences
2. Overlapping or contrasting concepts
3. Reasoning patterns likely required to solve each

Do not attempt to solve the new problem.
Begin your response with: 'Comparison:'
```

## C.5 USEFUL REASONING EXTRACTION PROMPT

This prompt distills essential reasoning from a detailed chain of thought, which is used to get a purified reasoning for further usage in method transfer.

**Useful Reasoning Extraction Prompt**

```
I have an example question:
{example_question}

And I have a detailed chain of thought (COT) for solving it:
{example_cot}

Your task:
1. Read the question and the detailed COT.
2. Extract only the essential steps needed to solve the question
    (the minimal reasoning path).
3. Omit any irrelevant or repetitive details, side explorations,
    or speculation.
4. Present the final answer clearly at the end.

Please provide a concise chain of thought that contains only the
    necessary logic and calculations
to solve the question, followed by the final answer.
```

### C.6 TEACHER PROMPT

This prompt extracts transferable strategies from an example solution.

**Teacher Prompt**

```
You are given one example question and one new question. A short
    comparison between them is also provided.
Example Question:
{example_question}
Example Solution:
{example_cot}
New Question:
{current_question}
{comparison}

Your Task:
Based on the comparison, find useful ideas or strategies in the
    example solution that can help solve the new question.

Instructions:
1. Only focus on strategies that are likely to transfer.
2. Skip those not relevant to the new question.
3. Do not try to solve the new question.
4. Just extract helpful techniques or methods from the example.

Begin your response with: 'Insights:'
```

### C.7 SUGGESTION PROMPT

This prompt asks the model to identify a single issue type and specify a concrete sanity check to perform.

```
Suggestion: Issue Identification & Sanity Check

You're an expert in the reasoning field. Below is:

**Question:**
{current_question}

**Current Best Reasoning Path:**
{best_path}

---

**Steps of your task:**
1. Identify **ONE** Issue Type (if any):
- Computational (math/units)
- Logical (reasoning gaps)
- Assumption (unverified premises)
- Interpretation (misaligned goals)

2. Perform Sanity Check:
- Computational: Recalculate or validate units/dimensions
- Logical: Test with edge cases or inverse reasoning
- Assumption: Explicitly list and challenge hidden premises
- Interpretation: Compare solution goals to problem verbatim

3. Your response should be of the compact JSON format:
   {"issue_and_sanity_check":"$CONTENT"} where CONTENT is the
   identified issue and corresponding sanity check you need to
   fill in.
```

## C.8 REVIEW PROMPT

This prompt evaluates multiple candidate checks and selects the best one to correct the reasoning.

```
Review: Select the Best Check

You are an expert reasoning evaluator. Below is:

**Question:**
{current_question}

**Current Reasoning Path:**
{best_path}

---

**Candidate Checks:**
{suggestion_prompt}

**Steps of your task:**
1. For each candidate check, evaluate whether applying its
   suggestion to the original reasoning would correct any mistake
   and lead to the correct final answer.
2. Choose the best check of the candidates in your opinion.
3. Your response should be of the compact JSON format:
   {"the_best_check":"$CONTENT"} where CONTENT is the details of
   the best check you need to fill in.
```

## C.9 REFINEMENT PROMPT

This prompt refines the reasoning path according to the chosen issue & sanity check, ensuring the final path is robust.

---

**Refinement: Fix Reasoning with Sanity Check**

```
You're an expert in the reasoning field. Below is:

**Question:**
{current_question}

**Current Best Reasoning Path:**
{best_path}

---

**Issue to Fix (with sanity check):**
{issue}

**Steps of your task:**
1. **Review** the issue and sanity check mentioned above.
2. **Refine** the reasoning path to fix given issue and ensure it
    passes the sanity check.
3. Before finalizing, figure out any potential problems with your
    approach and fix them step by step.
```

## D  LABELING PROTOCOL OF FAILURE-MODE STUDY

This prompt specifies the protocol for labeling failure cases by identifying how demonstrations influence model errors.

---

**Labeling Protocol of Failure-Mode Study**

```
You analyze how demonstrations influence model errors.
Your task: classify the error into one of three types and justify
    with evidence.
Error types:
1. semantic_misleading
   – Demo wording or concepts are copied/mirrored in the model's
       reasoning in an inappropriate way.
   – Surface overlap (terms, phrases, solution templates) steers
       the model away from the correct solution.
2. transfer_failure
   – The model tries to follow the demo's general strategy or
       structure, but applies it incorrectly.
   – The high-level approach is right, but details, calculations,
       or conditions are wrong.
3. other
   – The mistake would plausibly occur even without the demos.
   – No meaningful lexical, structural, or conceptual influence
       from the demos can be identified.
Use the following evidence channels:
– Lexical overlap: shared terms/concepts between demos and model
    reasoning.
– Structural similarity: similar step-by-step structure, solution
    template, or reasoning frame.
– Conceptual influence: demo concepts or focus areas incorrectly
    reused in the target reasoning.
– Direct references: explicit mentions of "example", "similar to
    above", etc.
Inputs:
– DEMONSTRATIONS: {demo_text}
– TARGET QUESTION: {target_question}
– MODEL'S REASONING: {model_cot}
– Correct answer: {correct_answer}
– Model answer: {model_answer}
Carefully inspect demos and model reasoning, then respond in JSON:
{
  "error_type": "semantic_misleading" | "transfer_failure" |
      "other",
  "lexical_overlap": ["list of overlapping terms or concepts"],
  "structural_similarity": "short description or 'none'",
  "conceptual_confusion": "short description or 'none'",
  "direct_references": "quotes or 'none'",
  "reasoning": "2-5 sentences explaining your decision with
      evidence",
  "confidence": "high" | "medium" | "low",
  "key_evidence": "1-2 sentences with the most important evidence",
  "demo_influence": "brief summary of how demos affected the error
      (or 'none')"
}
```

---

## E  LLM USAGE

We used large language models (ChatGPT and Gemini) as writing and formatting assistants. In particular, it helped refine grammar and phrasing, improve clarity, and suggest edits to figure/table captions and layout (e.g., column alignment, caption length, placement). The LLM did not con-

tribute to research ideation, experimental design, implementation, data analysis, or technical content beyond surface-level edits. All outputs were reviewed and edited by the authors, who take full responsibility for the final text and visuals.

