# OpenReview forum: "From Harm to Help: Turning Reasoning In-Context Demos into Assets for Reasoning LMs"
_ICLR.cc/2026/Conference — Submitted to ICLR 2026_

### Official Review · Reviewer_J6ds · 2025-10-20

**Soundness:** 3
**Presentation:** 2
**Contribution:** 3
**Rating:** 6
**Confidence:** 4

**Summary:**

The paper identifies two causes for the poor performance of few-shot CoT in reasoning LLMs—semantic misguidance and strategy transfer failure—and introduces Insight-to-Solve, a test-time framework that extracts transferable insights from demonstrations to guide reasoning. I2S and its refined variant I2S+ consistently improve performance across open- and closed-source models, including GPT-4.1.

**Strengths:**

- This paper comprehensively explore the underlying mechanisms behind the bad performance of few-shot CoT on RLMs, revealing the two mechanisms, semantic misguidance and strategy transfer failure.
- This paper proposes a novel framework Insight-to-Solve, a test-time framework that extracts transferable insights from demonstrations to guide reasoning for RLMs.
- I2S framework achieves performance improvements on GPQA and AIME across open-source models, and even achieves performance boost on AIME upon GPT4.1.

**Weaknesses:**

- The evaluation benchmarks are relatively narrow, and the GPT series models show substantial improvement on only one benchmark. More comprehensive experiments are needed to demonstrate the framework’s generalizability across diverse reasoning tasks.
- The study only reports results on Qwen3-1.7B, the smallest model in the Qwen3 series. Additional experiments on larger state-of-the-art reasoning LLMs are necessary to verify the scalability and robustness of the proposed framework.
- Presentation and typographical issues. In Figure 2, the numerical labels overlap with the histogram bins. Minor typos are also present, such as "expriment" (line 359) and "Closed-soruce" (line 99).

**Questions:**

- The authors should clarify why only the smallest model in the Qwen3 series (Qwen3-1.7B) was chosen for evaluation. Given that larger Qwen3 "thinking" models are available and well-suited for reasoning tasks, evaluating them would provide a more comprehensive validation of the framework.
- The reported results are averaged over five runs, but the paper does not include confidence intervals or statistical significance tests. The authors should provide these to assess the robustness and reliability of the observed improvements.

---

> ### Author Response · Authors · 2025-11-21
> **Rebuttal by Authors**
>
> We appreciate the reviewer’s encouraging feedback! We are glad that our work is seen as "comprehensively" exploring the underlying mechanisms behind the poor performance of few-shot CoT on RLMs. We address the reviewer’s remaining questions and concerns in detail below.
>
> ---
>
> **W1: The evaluation benchmarks are relatively narrow, and the GPT series models show substantial improvement on only one benchmark. More comprehensive experiments are needed to demonstrate the framework’s generalizability across diverse reasoning tasks.**
>
> Thank you for this suggestion. We would like to clarify our experimental design. AIME’25 is a math reasoning dataset that requires the model to generate a final numeric answer, while GPQA Diamond is a multiple-choice benchmark, making it structurally different from AIME’25. The General Reasoning – Eng and General Reasoning – General suites, in contrast, consist of open-ended reasoning tasks, which further distinguish them from both AIME’25 and GPQA.
>
> Taken together, these four settings already cover diverse reasoning regimes: math vs. non-math, open-ended generation vs. multiple-choice. Given the limited computational budget during the rebuttal period, we chose to prioritize deeper analysis rather than adding new benchmarks. In the revision, $\\textrm{\\color{blue}Section 5.5}$ now discusses the compute–performance trade-off, and $\\textrm{\\color{blue}Section 5.6}$ provides an in-depth study of the failure-mode distribution.
>
>
> ---
>
> **W2: The study only reports results on Qwen3-1.7B, the smallest model in the Qwen3 series. Additional experiments on larger state-of-the-art reasoning LLMs are necessary to verify the scalability and robustness of the proposed framework.**
>
> Thank you for the suggestion. We have added experiments with **Qwen3-14B** to further validate our method. The revised manuscript shows that I2S/I2S+ provide consistent gains for this larger model as well.
>
> ---
>
> **W3: Presentation and typographical issues. In Figure 2, the numerical labels overlap with the histogram bins. Minor typos are also present, such as "expriment" (line 359) and "Closed-soruce" (line 99).**
>
>
> We appreciate the detailed eye for presentation! We have fixed it in revision.
>
> ---
>
> **Q1: The authors should clarify why only the smallest model in the Qwen3 series (Qwen3-1.7B) was chosen for evaluation. Given that larger Qwen3 "thinking" models are available and well-suited for reasoning tasks, evaluating them would provide a more comprehensive validation of the framework.**‘
>
>
> Our initial choice of Qwen3-1.7B was primarily constrained by the limited GPU resources available on our school-level cluster before the conference deadline: larger Qwen3 models are substantially more expensive to evaluate across all benchmarks and ablations. Following your suggestion, we have now included results for **Qwen3-14B** in revision.
>
>
> ---
>
> **Q2: The reported results are averaged over five runs, but the paper does not include confidence intervals or statistical significance tests. The authors should provide these to assess the robustness and reliability of the observed improvements.**
>
> Thank you for pointing this out. In the revised version, we include a table reporting the standard deviation over five runs for all methods and benchmarks ($\\textrm{\\color{blue}Appendix B.5}$).

---

> > ### Comment · Reviewer_J6ds · 2025-11-26
> >
> > Thank you for the authors’ detailed responses. While most of my concerns have been adequately addressed, I remain concerned about the limited scope of the evaluation benchmarks presented in the paper. Therefore, I will maintain my original score. Nonetheless, I am still in favor of accepting this paper.

---

> > > ### Author Response · Authors · 2025-11-27
> > > **Appreciation for the Reviewer’s Follow-Up and Support**
> > >
> > > Thank you again for your thoughtful feedback and for carefully considering our rebuttal. We appreciate your support for accepting the paper!

---

### Official Review · Reviewer_v1vB · 2025-10-30

**Soundness:** 2
**Presentation:** 3
**Contribution:** 2
**Rating:** 4
**Confidence:** 4

**Summary:**

The paper shows that few-shot Chain-of-Thought (CoT) demos can hurt LLMs because (i) semantic misleading makes models copy surface patterns from similar examples, and (ii) strategy-transfer failure prevents extracting and applying useful tactics to the target problem. Building on this diagnosis, the authors propose I2S (Insight-to-Solve): a sequential test-time pipeline that (a) compares target vs. demo to extract reusable insights, (b) generates a target-specific reasoning trace from those insights (not the raw demo), and (c) decouples final answering from the demos to avoid copying; I2S+ adds brief self-refinement of the trace. Across AIME’25 and GPQA and multiple models (open and closed), I2S/I2S+ outperforms direct answering.

**Strengths:**

The paper is clearly written and tells a compelling story, with strong narrative flow from motivation to method. The overall presentation is smooth and coherent, and the method design feels tightly connected to the empirical findings.

The paper works on a counterintuitive yet important question: why ICL hurts reasoning. The findings are insightful and also drive the proposed method.

The I2S can also effectively leverage extra compute to further improve the reasoning performance.

**Weaknesses:**

The study focuses primarily on **one-shot ICL**, which overlooks the essence of in-context learning—its strength in the **many-shot** regime where richer exemplar context is available. This limitation makes the analysis and claims feel overstated.

The section explaining why ICL hurts reasoning is not fully convincing. It relies on two qualitative observations without sufficient quantitative evidence at scale, making it unclear whether these factors truly explain the degradation in performance. The examples are also one-shot, which may not generalize to many-shot cases.

The proposed method’s scalability to **many-shot settings** remains questionable. It is unclear how the approach would effectively fuse or aggregate information from multiple exemplars, which is crucial for practical deployment in real-world ICL scenarios.

Section 4.2 is a bit hard to follow since it does not explicitly show how each component is done in details. Explicitly referring to the prompts used can help. At present, the writing seems to obscure these details, which makes it harder for the reader to understand and evaluate the core mechanics of the approach.

**Questions:**

Why not use model’s self-generated reasoning paths as in-context exemplars? Reasoning generated by a different teacher model may not be the best exemplar for the student to learn.

---

> ### Author Response · Authors · 2025-11-21
> **Rebuttal by Authors [1/2]**
>
> We thank the reviewer for the thoughtful feedback. We appreciate the reviewer’s encouraging feedback! We also appreciate the recognition that we tackle a "counterintuitive yet important question: why ICL hurts reasoning", and that our findings are "insightful" and drive the proposed method to further improve performance. We address the remaining concerns below.
>
> ---
>
> **W1: The study focuses primarily on one-shot ICL, which overlooks the essence of in-context learning—its strength in the many-shot regime where richer exemplar context is available. This limitation makes the analysis and claims feel overstated.**
>
> We appreciate this perspective and the emphasis on many-shot in-context learning as an important regime. However, we respectfully do not fully agree with the general claim that “the essence” of ICL lies exclusively in the many-shot setting—especially for reasoning models.
>
> For earlier non-reasoning LMs, it is indeed widely observed that performance often improves as more exemplars are added. For reasoning models, whether this trend still holds and should hold is, to our knowledge, an open question. In our work, we empirically show that, for such models, adding more demonstrations can systematically hurt performance, regardless of whether we use “thinking” demonstrations (Q + reasoning + A) or shorter “non-thinking” demonstrations (Q + A). These results are summarized in our newly added $\\textrm{\\color{blue}Appendix B.1}$.
>
> You also raise, in Weakness 3, a related concern: “The proposed method’s scalability to many-shot settings remains questionable. It is unclear how the approach would effectively fuse or aggregate information from multiple exemplars, which is crucial for practical deployment in real-world ICL scenarios.”
>
> Although we do not believe that extending to higher shot counts is conceptually central to our work’s main message, we agree that this is an important concern and have addressed it carefully. To this end, we have added new experiments directly applying I2S to a practical model, GPT-4.1, to study its "practical deployment" with up to 3 demonstrations. On AIME and GPQA, we compare direct prompting with I2S in 1-shot, 2-shot, and 3-shot settings. The results show that:
>
>
>
> | Dataset (GPT-4.1) | direct QA     | I2S 1-shot        | I2S 2-shot        | I2S 3-shot        |
> | :---------------- | :-----------: | :---------------: | :---------------: | :---------------: |
> | AIME              | 34.00±4.90    | **48.00±4.52**    | 43.33±5.27        | 47.33±3.56        |
> | GPQA              | 67.98±1.88    | 68.50±0.87        | 68.29±0.75        | **69.07±0.89**    |
>
>
> On AIME, I2S 1-shot substantially improves over direct prompting, while 2-shot and 3-shot variants provide comparable but not consistently higher gains.
>
> On GPQA, I2S 1-shot again improves over direct prompting, and 2–3 shots yield similar or slightly improved accuracy.
>
> Overall, these results indicate that our method does extend to the few-shot regim on practical model GPT-4.1.
>
>
> ---
>
> **W2: The section explaining why ICL hurts reasoning is not fully convincing. It relies on two qualitative observations without sufficient quantitative evidence at scale, making it unclear whether these factors truly explain the degradation in performance. The examples are also one-shot, which may not generalize to many-shot cases.**
>
> We appreciate this concern. To strengthen the diagnosis, we have added the quantitative failure-mode analysis in $\\textrm{\\color{blue}Section 5.6}$ of revision:
>
> Under in-context reasoning, semantic misleading alone accounts for a large fraction of questions (e.g., 48.6% on AIME'25 / R1-Distill-Qwen-7B and 43.3% on GPQA / Qwen3-1.7B). With our method, these fractions drop consistently across models and datasets, providing quantitative evidence that the two failure modes identified in $\\textrm{\\color{blue}Section 3}$ explain a substantial portion of the performance degradation. For more details, please check $\\textrm{\\color{blue}Figure 7}$.
>
> ---
>
> **W4: Section 4.2 is a bit hard to follow since it does not explicitly show how each component is done in details. **Explicitly referring to the prompts used can help.** At present, the writing seems to obscure these details, which makes it harder for the reader to understand and evaluate the core mechanics of the approach.**
>
> We appreciate your suggestions on the writing, Thanks! We update the reference correspondly in $\\textrm{\\color{blue}Section 4.2}$.

---

> ### Author Response · Authors · 2025-11-21
> **Rebuttal by Authors [2/2]**
>
> ---
>
> **Q1: Why not use model’s self-generated reasoning paths as in-context exemplars? Reasoning generated by a different teacher model may not be the best exemplar for the student to learn.**
>
> **Our work is motivated by a puzzling observation:** the same training samples used to distill R1-Distill-Qwen-7B, when reused as few-shot reasoning demonstrations, can noticeably decrease its performance. This behavior is at odds with the common view of in-context learning as a gradient-descent-like mechanism on those examples [1,2,3]. **Then, naturally,we focus on diagnosing why these reasoning demonstrations can backfire** and then on designing a method that efficiently leverages them at test time.
> Exploring whether a model’s own self-generated reasoning could serve as better in-context exemplars is an interesting direction, but obviously it addresses a different question and is beyond the scope of the present work.
>
>
> ---
>
> ***References***
>
> [1] Dai, Damai, et al. "Why can GPT learn in-context? language models secretly perform gradient descent as meta-optimizers." Findings of the Association for Computational Linguistics: ACL 2023. 2023.
>
> [2] Ahn, Kwangjun, et al. "Transformers learn to implement preconditioned gradient descent for in-context learning." Advances in Neural Information Processing Systems 36 (2023): 45614-45650.
>
> [3] Von Oswald, Johannes, et al. "Transformers learn in-context by gradient descent." International Conference on Machine Learning. PMLR, 2023.

---

> ### Comment · Reviewer_v1vB · 2025-11-25
>
> Thank you for your response on the many-shot ICL and quantitative analysis of the failure mode. The updated version is much better. I have increased my score.

---

> > ### Author Response · Authors · 2025-11-25
> > **Thank you for your feedback and improved assessment**
> >
> > Thank you for your time and constructive suggestions. We’re happy to hear that the updated version is much better and really appreciate your improved assessment!

---

### Official Review · Reviewer_u8De · 2025-10-30

**Soundness:** 3
**Presentation:** 3
**Contribution:** 2
**Rating:** 4
**Confidence:** 4

**Summary:**

The paper examines why few-shot reasoning demonstrations can harm verifier-trained reasoning LLMs. It identifies two causes, semantic misleading and strategy transfer failure, that lead models to copy rather than reason. To address this, the authors propose Insight-to-Solve (I2S), which turns demonstrations into explicit, reusable insights, and I2S+, which refines reasoning for coherence. Experiments on AIME’25, GPQA, and other benchmarks show consistent accuracy gains and shorter reasoning traces, reframing demonstrations as sources of transferable insights rather than direct exemplars.

**Strengths:**

- The paper proposed I2S and I2S+ frameworks, which are conceptually simple yet effective, converting demonstrations into reusable reasoning insights and optionally refining reasoning for coherence.
- Results show consistent accuracy improvements across multiple reasoning benchmarks while reducing reasoning length by about 20%.

**Weaknesses:**

- The technical novelty of I2S is limited, which is just prompting engineering. The method essentially formalizes a multi-step prompting or reasoning refinement pipeline, similar to prior “deliberation,” “reflect,” or “meta-CoT” approaches, without introducing new learning objectives, model architectures, or theoretical grounding.
- I2S relies on additional model calls, increasing test-time latency, yet there is no discussion of compute–accuracy trade-offs or efficiency at scale.
- The paper identifies two failure modes, semantic misleading and strategy-transfer failure,  but never quantifies their frequency or shows examples where I2S still fails.
- ICLR 2025 -> ICLR 2026

**Questions:**

- How sensitive are the results to the phrasing of the insight-extraction and refinement prompts? Have the authors evaluated prompt robustness or tried automatic prompt tuning?
- Since I2S and I2S+ require multiple model calls per query, could the authors report the average inference latency or total token usage to substantiate the claim that compute cost is comparable to few-shot CoT?
- The paper identifies semantic misleading and strategy-transfer failure as two error modes. Could the authors provide quantitative statistics or examples showing where I2S still fails?
- The context-length analysis (App. B.2, Table 5) shows reasoning demonstrations reaching up to 13–14 k tokens. How does context truncation affect performance, and could shorter exemplars achieve similar gains?
- Is there any theoretical or empirical evidence that “insight extraction” leads to more faithful intermediate reasoning (e.g., fewer contradictions or hallucinations) compared to standard CoT?

---

> ### Author Response · Authors · 2025-11-21
> **Rebuttal by Authors [1/3]**
>
> We appreciate the reviewer’s encouraging feedback! And we are glad that the our framework are viewed as "conceptually simple yet effective". We respond to the remaining comments in detail below.
>
> ---
>
> **W1: The technical novelty of I2S is limited, which is just prompting engineering. The method essentially formalizes a multi-step prompting or reasoning refinement pipeline, similar to prior “deliberation,” “reflect,” or “meta-CoT” approaches, without introducing new learning objectives, model architectures, or theoretical grounding.**
>
>
> We appreciate this perspective. I2S, as a test-time scaling method, operates entirely at test time, without introducing new training objectives or architectures. Our goal in this work is primarily empirical: to diagnose and mitigate the regimes where reasoning demonstrations hurt verifier-trained reasoning models, rather than to propose a new training algorithm.
>
> **Our motivation comes from a puzzling observation:** the very training samples used to distill R1-Distill-Qwen-7B, when reused as few-shot reasoning demonstrations, can noticeably decrease its performance. This behavior is at odds with the common view of in-context learning as a gradient-descent-like mechanism operating on those same examples [1,2,3]. **Naturally, this motivates us to carefully analyze why such failures occur and to propose a method** that allows the model to benefit from, rather than be harmed by, reasoning demonstrations. To our knowledge, this phenomenon—and how to address it—is not explored in prior “deliberation,” “reflect,” or “meta-CoT” style works.
>
> Within this setting, our contribution is twofold. First, we provide an empirical diagnosis of this puzzling observation, identifying and quantitatively validating two dominant failure modes: semantic misleading and strategy-transfer failure. In $\\textrm{\\color{blue}Section 5.6}$ of the revision, we further quantitatively measure the frequency of these failure modes. Second, I2S is designed specifically around this diagnosis: it forces demonstrations to be abstract insights and decouples final answering from the raw demo trace, directly targeting the identified failure modes. This design is conceptually distinct from standard self-refinement pipelines that repeatedly revise a single CoT chain.
>
> ---
>
> **W2: I2S relies on additional model calls, increasing test-time latency, yet there is no discussion of compute–accuracy trade-offs or efficiency at scale.**
>
> We fully agree. For any test-time scaling method, the key is how much additional computation you trade for the corresponding performance gain. In the revision, we have added an analysis of the compute–accuracy trade-off ($\\textrm{\\color{blue}Section 5.5}$). For each model and dataset, we measure the average number of generated tokens per question over five runs and plot accuracy as a function of this generated-token budget for all methods, including one-shot CoT, Self-Refine, Progressive-Hint Prompting (PHP).
>
> We use the number of generated tokens as a proxy for computation cost because, in vLLM, the prefill phase (processing the input prompt) is highly optimized and relatively fast, while most of the wall-clock time is spent in autoregressive decoding. Consequently, the total number of generated tokens closely tracks both latency and FLOPs at test time. Across R1-Distill-Qwen-7B and Qwen3-14B on AIME'25 and GPQA, I2S consistently lies on a more favorable computation–performance frontier than few-shot CoT and other baselines. In other wods, for a fixed token budget, it matches or outperforms competing methods in accuracy, and to reach a given accuracy level, it requires fewer generated tokens.

---

> ### Author Response · Authors · 2025-11-21
> **Rebuttal by Authors [2/3]**
>
> ---
>
> **W3: The paper identifies two failure modes, semantic misleading and strategy-transfer failure, but never quantifies their frequency or shows examples where I2S still fails.**
>
> We agree that this was under-specified in the original version. In the revision, we add a failure-mode study ($\\textrm{\\color{blue}Section 5.6}$), where we use GPT to annotate errors on AIME'25 and GPQA into three categories: semantic misleading, strategy-transfer failure, and other errors, including calculation mistakes and genuinely incorrect internal reasoning (i.e., hard questions that are simply beyond the capability of this model).
>
> Across both R1-Distill-Qwen-7B and Qwen3-1.7B, we find that semantic misleading is the dominant error mode under one-shot prompting (*roughly 30–50% of failures*), with strategy-transfer failures forming a smaller but non-trivial portion. Under I2S, the share of both semantic- and strategy-transfer failures consistently drops, with most residual errors shifting into the heterogeneous "other", indicating that I2S largely neutralizes the harmful influence of demonstrations and leaves a residual set of more standard, heterogeneous reasoning mistakes instead.
>
> ---
>
> **W4: ICLR 2025 -> ICLR 2026**
>
> Thanks for pointing this out! We fixed it in revision.
>
> ---
>
> **Q1: How sensitive are the results to the phrasing of the insight-extraction and refinement prompts? Have the authors evaluated prompt robustness or tried automatic prompt tuning?**
>
> We did not perform prompt optimization or automatic prompt tuning. In line with our goal of keeping I2S simple and practical, we designed a set of straightforward, directive-style prompts (for insight extraction, target reasoning, and optional refinement) and then fixed these templates for expriments with all models and datasets. The fact that this single prompt set consistently improves performance across R1-Distill-Qwen-7B, Qwen3-1.7B, Qwen3-14B, GPT-4.1, and across AIME'25, GPQA, and open-ended tasks already suggests that I2S is not overly sensitive to phrasing.
>
> ---
>
> **Q2: Since I2S and I2S+ require multiple model calls per query, could the authors report the average inference latency or total token usage to substantiate the claim that compute cost is comparable to few-shot CoT?**
>
> Yes. For the Computation-Performance tradeoff, please check our newly added  $\\textrm{\\color{blue}Section 5.5}$  and our reply to W2.
>
> ---
>
> **Q3: The paper identifies semantic misleading and strategy-transfer failure as two error modes. Could the authors provide quantitative statistics or examples showing where I2S still fails?**
>
>
> We agree that quantifying failure modes is important. In the revision, we add a failure-mode study in $\\textrm{\\color{blue}Section 5.6}$, where we manually annotate model errors on AIME'25 and GPQA into three categories: (i) *semantic misleading*, (ii) *strategy transfer failure*, and (iii) *other errors*, including calculation mistakes and genuinely hard questions that are beyond the model’s capability. We conduct this analysis for both R1-Distill-Qwen-7B and Qwen3-1.7B (the two main models in our tables), comparing one-shot in-context reasoning and I2S.
> As shown in the newly added $\\textrm{\\color{blue}Figure 7}$, across all four (model, dataset) settings, *semantic misleading* is by far the dominant error mode under in-context reasoning, accounting for roughly 30–50% of all failures, while strategy-transfer failures contribute a smaller but non-negligible share. With I2S, the share of semantic-misleading errors consistently drops by about 15–30 percentage points and strategy-transfer failures also decrease, with most residual errors shifting into the heterogeneous "other" category. This shows that the two identified failure modes are quantitatively significant and that I2S directly mitigates them, while leaving a residual set of more standard reasoning and calculation mistakes. We also add a description of the labeling protocol in $\\textrm{\\color{blue}Appendix D}$.

---

> ### Author Response · Authors · 2025-11-21
> **Rebuttal by Authors [3/3]**
>
> ---
>
> **Q4: The context-length analysis (App. B.2, Table 5) shows reasoning demonstrations reaching up to 13–14 k tokens. How does context truncation affect performance, and could shorter exemplars achieve similar gains?**
>
> Thanks for your careful reading. As we add more long reasoning demonstrations, they gradually consume the available context. When with a sufficiently large number of demonstrations used as ICL examples, the model’s performance will drop quickly because there is no longer enough remaining space for the model to "think" about the new target question.
>
> For the second part of the question ("could shorter exemplars achieve similar gains?"), we interpret this as asking whether the same degradation trend appears even when demonstrations are short.
>
> To examine this, we also provide ICL experiments with Q+A demonstrations (without the reasoning part). These exemplars are much shorter and do not heavily consume the context budget (note that our original setup uses Q+reasoning+A). We observe the same trend as in $\\textrm{\\color{blue}Figure 1}$: as the number of demonstrations increases, the performance consistently drops. We include this $\\textrm{\\color{blue}Figure 8}$ and analysis in the newly added $\\textrm{\\color{blue}Appendix B.1}$.
>
> ---
>
> **Q5: Is there any theoretical or empirical evidence that “insight extraction” leads to more faithful intermediate reasoning (e.g., fewer contradictions or hallucinations) compared to standard CoT?**
>
> We agree that this is an important question. While a full theoretical guarantee is beyond our scope, we do provide empirical evidence. In $\\textrm{\\color{blue}Section 5.6}$, we add a failure-mode study that quantitatively measures how the error modes change after applying our method, which directly addresses to what extent I2S leads to more faithful intermediate reasoning.
>
> $\\textrm{\\color{blue}Section 5.6}$ presents a dedicated failure-mode study, where we manually annotate model errors on AIME'25 and GPQA into three categories: (i) semantic misleading, (ii) strategy-transfer failure, and (iii) other errors, including calculation mistakes and genuinely hard questions that are beyond the model’s capability. We conduct this analysis for both R1-Distill-Qwen-7B and Qwen3-1.7B (the two main models in our tables), comparing one-shot in-context reasoning with I2S.
> As shown in the newly added $\\textrm{\\color{blue}Figure 7}$, across all four (model, dataset) settings, semantic misleading is by far the dominant error mode under in-context reasoning, accounting for roughly 30–50% of all failures, while strategy-transfer failures contribute a smaller but non-negligible share. With I2S, the proportion of semantic-misleading errors consistently drops by about 15–30 percentage points, and strategy-transfer failures also decrease, with most residual errors shifting into the heterogeneous "other" category. This indicates that the two identified failure modes are quantitatively significant and that I2S directly mitigates them, leaving a residual set of more standard reasoning and calculation mistakes. We also add a description of the labeling protocol in $\\textrm{\\color{blue}Appendix D}$.
>
> ---
>
> ***References***
>
> [1] Dai, Damai, et al. "Why can GPT learn in-context? language models secretly perform gradient descent as meta-optimizers." Findings of the Association for Computational Linguistics: ACL 2023. 2023.
>
> [2] Ahn, Kwangjun, et al. "Transformers learn to implement preconditioned gradient descent for in-context learning." Advances in Neural Information Processing Systems 36 (2023): 45614-45650.
>
> [3] Von Oswald, Johannes, et al. "Transformers learn in-context by gradient descent." International Conference on Machine Learning. PMLR, 2023.

---

> ### Comment · Reviewer_u8De · 2025-11-26
>
> Thanks for the authors' efforts.

---

> > ### Author Response · Authors · 2025-11-27
> > **Follow-up on Our Responses to Your Review**
> >
> > Thank you again for taking the time to read our paper and for your earlier detailed comments and questions! We also appreciate your follow-up note: "Thanks for the authors’ efforts."
> >
> > Since you raised several important concerns (Q1–Q5) and we provided point-by-point responses and new analyses/experiments in the rebuttal, we wanted to kindly check whether our replies have adequately addressed your main concerns. From your latest message alone, it is a bit hard for us to tell whether you now consider these issues resolved, especially as the overall score has not changed.
> >
> > If some of your concerns are still unresolved or if any of our clarifications were unclear, we would be very grateful if you could briefly let us know!

---

> > > ### Comment · Reviewer_u8De · 2025-11-27
> > >
> > > Thanks for the additional experiments, explanations, and cases in the updated paper. These addressed my concerns. I will raise the score.

---

> > > > ### Author Response · Authors · 2025-11-27
> > > > **Thank You for Your Timely Follow-Up**
> > > >
> > > > Many thanks for your timely follow-up! We really appreciate your careful review and feedback.

---

### Official Review · Reviewer_EWz4 · 2025-10-31

**Soundness:** 2
**Presentation:** 3
**Contribution:** 2
**Rating:** 4
**Confidence:** 3

**Summary:**

In-context learning with few shot demonstrations was the standard method to improve accuracy of LLM. However, authors found that this method is not applicable to large reasoning models (LRMs), which is already trained to reason with post training using verifier based RL (such as qwen3 and deepseek-r1). The authors identified two failure scenario: (i) semantic misguidance, where high textual similarity leads the model to treat the target as the same as the demonstrations and to copy intermediate steps verbatim; and (ii) strategy transfer failure, where the model is unable to find reasoning strategy and apply them to target question.

To solve this, author proposed Insight-to-Solve (I2S) Framework, which turns demonstrations into insights that is reuseable and instance specific. Specifically, they first extracts abstract insights from demonstrations. Then applies them to the target question to generate a reasoning trace. The 3rd step is optional, which performs self-refinement for coherence and correctness (I2S+). In the final step, they decouples final answer generation from the demonstrations. They find that I2S outperform direct answering and I2S+ outperform other baselines (e.g., Self-Refine, Progressive-Hint Prompting) across Closed-ended benchmarks (AIME’25 and GPQA) and Open-ended tasks (GeneralReasoning Engineering and General domains).

**Strengths:**

The paper is well-written and logically organized, making complex ideas accessible without oversimplifying.

The paper introduces a novel framework Insight-to-Solve that reimagines how reasoning demonstrations are used at inference time.

**Weaknesses:**

How common are the two failure scenario? While the paper identifies two key failure scenario (semantic misguidance and strategy transfer failure), the analysis is mostly qualitative and lacks quantitative evidence. Without quantification, it’s hard to know: Are these dominant failure modes? Are there other important categories?

Why two failure scenario happen? The framework is empirically motivated, but lacks a formal theoretical explanation of why these failure scenario happens. Could the root cause be: cross-attention shortcut completion? or inductive biases toward surface similarity?

What is the computation cost of I2S when compared to baselines? I2S introduces multi-step inference and I2S+ further introduce refinement, which makes it much more costly and lower latency, when compared to baselines such as self-refine and Progressive-Hint Prompting. Could author provide a performance plot where y axis is the accuracy and x axis is the cost (number of input/output tokens)

How good is the I2S on solving the strategy transfer failure?  Authors show the I2S performance on multiple reasoning datasets to test its reasoning ability, while the paper claims that I2S fixes strategy-transfer failure in section 4. But the motivation on strategy transfer failure is not yet tested. Existing datasets (AIME, GPQA) are single-instance reasoning, not strategy-pair tasks, and not designed to probe transfer. Can authors test their method on reasoning strategy dataset that measures strategy transfer, such as [NaturalThoughts](https://arxiv.org/html/2507.01921v1), Abstraction and Reasoning Corpus, or [ARB](https://arb.duckai.org/). ARC provides gold standard for strategy transfer: latent transformation rules and distractor patterns. ARB splits tasks into multiple subcategory: Law, MCAT Science, Physicis numerical and etc for authors to transfer.

**Questions:**

Can authors share a list of transferable strategies? This can help readers understand what author actually meant

---

> ### Author Response · Authors · 2025-11-21
> **Rebuttal by Authors [1/3]**
>
> We thank the reviewer for the thoughtful feedback and appreciate the recognition of Insight-to-Solve as a "novel" framework that reimagines how reasoning demonstrations are used at test time. We address the remaining questions and concerns below.
>
> ---
>
> **W1: How common are the two failure scenario? While the paper identifies two key failure scenario (semantic misguidance and strategy transfer failure), the analysis is mostly qualitative and lacks quantitative evidence. Without quantification, it’s hard to know: Are these dominant failure modes? Are there other important categories?**
>
> We agree that quantifying failure modes is important. In the revision, we **add a failure-mode study**  in $\\textrm{\\color{blue}Section 5.6}$, where we use GPT to annotate model errors on AIME'25 and GPQA into three categories: (i) *semantic misleading*, (ii) *strategy transfer failure*, and (iii) *other errors*, including calculation mistakes and genuinely hard questions that are beyond the model’s capability. We conduct this analysis for both R1-Distill-Qwen-7B and Qwen3-1.7B (the two main models in our tables), comparing one-shot in-context reasoning and I2S.
> As shown in the newly added $\\textrm{\\color{blue}Figure 7}$, across all four (model, dataset) settings, *semantic misleading* is by far the dominant error mode under in-context reasoning, accounting for roughly 30–50\% of all failures, while strategy-transfer failures contribute a smaller but non-negligible share. With I2S, the share of semantic-misleading errors consistently drops by about 15–30% and strategy-transfer failures also decrease, with most residual errors shifting into the heterogeneous "other" category. This shows that the two identified failure modes are quantitatively significant, and that I2S largely **neutralizes their harmful influence**, leaving a residual set of more standard reasoning and calculation mistakes. We also add a description of the labeling protocol in $\\textrm{\\color{blue}Appendix D}$.
>
>
> ---
>
> **W2: Why two failure scenario happen? The framework is empirically motivated, but lacks a formal theoretical explanation of why these failure scenario happens. Could the root cause be: cross-attention shortcut completion? or inductive biases toward surface similarity?**
>
> We appreciate this forward-looking question. Our goal in this work is primarily empirical: to diagnose and mitigate the regimes in which reasoning demonstrations hurt large reasoning models (LRMs). **Our motivation comes from a puzzling observation:** the training samples used to distill R1-Distill-Qwen-7B, when inserted back as in-context demonstrations, can noticeably decrease its performance, which seems at odds with the common view of in-context learning as a kind of gradient descent [1,2,3]. A full theoretical account of this phenomenon is beyond our current scope, but we now include a brief mechanistic discussion:
>
> > **For semantic misleading:** In LRMs, demonstrations and target live in a shared context, and strong cross-attention to highly similar tokens encourages *pattern completion*: the model continues the most probable continuation of the demo-like sequence rather than solving the new instance. This is consistent with prior observations that LLMs can over-weight surface similarity when reading multiple QA pairs in a single prompt [4,5].
>
> > **For strategy transfer failure**: Reasoning strategies (e.g., "set up equations before plugging in numbers") are abstract and require *inductive transfer* rather than copying. When demonstrations are long, the model might focus on local token-level patterns instead of extracting high-level invariants, leading to failure to apply the correct strategy to the target.
>
> We now make this discussion clear in $\\textrm{\\color{blue}Section 3.2}$ of revision and explicitly acknowledge that a more formal analysis is an exciting direction for future work.

---

> ### Author Response · Authors · 2025-11-21
> **Rebuttal by Authors [2/3]**
>
> ---
>
> **W3: What is the computation cost of I2S when compared to baselines? I2S introduces multi-step inference and I2S+ further introduce refinement, which makes it much more costly and lower latency, when compared to baselines such as self-refine and Progressive-Hint Prompting. Could author provide a performance plot where y axis is the accuracy and x axis is the cost**
>
> We fully agree and have **added a new subsection on compute–accuracy trade-offs** ($\\textrm{\\color{blue}Section 5.5}$). For each model and dataset, we measure the average number of generated tokens per question over five runs and plot accuracy vs. generated-token budget for all methods, including one-shot CoT, Self-Refine, Progressive-Hint Prompting (PHP).
>
> We use the number of generated tokens as a proxy for computation cost because, in vLLM, the prefill phase (processing the input prompt) is highly optimized and relatively fast, while most of the wall-clock time is spent in autoregressive decoding. Consequently, the total number of generated tokens closely tracks both latency and FLOPs at test time. Across R1-Distill-Qwen-7B and Qwen3-14B on AIME'25 and GPQA, I2S consistently lies on a more favorable computation–performance frontier than few-shot CoT and other baselines. In other words, for a fixed token budget, I2S matches or outperforms competing methods in accuracy; to reach a given accuracy level, our method requires fewer generated tokens.
>
> ---
>
> **W4: How good is the I2S on solving the strategy transfer failure? Authors show the I2S performance on multiple reasoning datasets to test its reasoning ability, while the paper claims that I2S fixes strategy-transfer failure in section 4. But the motivation on strategy transfer failure is not yet tested. Existing datasets (AIME, GPQA) are single-instance reasoning, not strategy-pair tasks, and not designed to probe transfer. Can authors test their method on reasoning strategy dataset that measures strategy transfer, such as NaturalThoughts, Abstraction and Reasoning Corpus, or ARB. ARC provides gold standard for strategy transfer: latent transformation rules and distractor patterns. ARB splits tasks into multiple subcategory: Law, MCAT Science, Physicis numerical and etc for authors to transfer.**
>
>
>
> Thank you for these suggestions. Before addressing how well I2S tackles strategy-transfer failures, we would like to clarify our motivation and what "strategy transfer" means in our setting.
>
> Our work is motivated by a puzzling empirical observation: the training samples used to distill R1-Distill-Qwen-7B, when re-used as few-shot reasoning demonstrations, can decrease its performance. This behavior contrasts with the common view of in-context learning as a gradient-descent-like mechanism on those in-context examples.
>
> While AIME'25 and GPQA are indeed single-instance benchmarks by design, our experiments follow the demonstration–target pair protocol. Specifically, for every test question we retrieve a separate reasoning demo (question + thinking + answer) from large reasoning data set (OpenThoughts-114K and GeneralThought-430K; $\\textrm{\\color{blue}Section 5.1}$). The model is then asked to leverage this demo when solving the target. *In this sense, every test case in our setup is already a strategy-transfer scenario.*
>
> Within this setting, our new *failure-mode study* ($\\textrm{\\color{blue}Figure 7}$) provides quantitative evidence that I2S improves strategy transfer. Under one-shot reasoning demo prompt, a non-trivial portion of errors are labeled as strategy transfer failures; with I2S, the fraction of such errors consistently decreases across both models and both datasets, indicating that the model is more often able to extract and apply the right strategy instead of ignoring or misusing the demo.
>
> We agree that benchmarks such as NaturalThoughts, ARC, and ARB are excellent probes for more abstract forms of strategy transfer, and we view them as highly complementary to our current focus on math/science-style reasoning traces for R1-style LRMs.

---

> ### Author Response · Authors · 2025-11-21
> **Rebuttal by Authors [3/3]**
>
> ---
>
> **Q1: Can authors share a list of transferable strategies? This can help readers understand what author actually meant**
>
> Thank you for the suggestion. We realize our setting may not have made this aspect fully clear. In our experiments, transferable strategies are instance-specific: each target question is paired with its own retrieved reasoning demo, so the concrete strategy to be transferred depends on that particular (demo, target) pair rather than a single global list shared across all questions (as discussed in our response to W4).
>
> Although there is no single set of universal transferable strategies, providing examples can indeed help clarify what we mean. In $\\textrm{\\color{blue}Figure 3}$, the green-highlighted sentence is an extracted transferable insight: "enumerate adjacent pairs" as a general problem-solving strategy, which can be applied to both the demonstration and the target question. We additionally highlight such transferable insights in $\\textrm{\\color{blue}Appendix B.3}$.
>
>
> ---
>
> ***References:***
>
> [1] Dai, Damai, et al. "Why can GPT learn in-context? language models secretly perform gradient descent as meta-optimizers." Findings of the Association for Computational Linguistics: ACL 2023. 2023.
>
> [2] Ahn, Kwangjun, et al. "Transformers learn to implement preconditioned gradient descent for in-context learning." Advances in Neural Information Processing Systems 36 (2023): 45614-45650.
>
> [3] Von Oswald, Johannes, et al. "Transformers learn in-context by gradient descent." International Conference on Machine Learning. PMLR, 2023.
>
> [4] Ali, Ameen, Lior Wolf, and Ivan Titov. "Mitigating copy bias in in-context learning through neuron pruning." arXiv preprint arXiv:2410.01288 (2024).
>
> [5] Gupta, Karan, et al. "How robust are LLMs to in-context majority label bias?." arXiv preprint arXiv:2312.16549 (2023).

---

### Author Response · Authors · 2025-12-03
**Summary of Rebuttal Revisions and General Response**

Dear Reviewers and Area Chairs,

Thank you for taking the time to review our paper! We are glad that the reviewers recognized the importance of our central question, that the very training samples used to train LRMs (e.g., R1-Distill-Qwen-7B) can hurt performance when reused as in-context demonstrations, and found our framework an effective test-time solution to this issue. After we added new experiments (e.g., failure-mode study, computation–performance trade-off analysis, larger-model results on Qwen3-14B, and a mechanistic discussion), two reviewers (u8De, v1vB) explicitly raised their overall scores from 4 to 6.

Below we summarize the main concerns and what we addressed in the rebuttal and revision:


| Reviewer | Main concerns (very brief) | What we did in rebuttal / revision |
| :--- | :--- | :--- |
| **EWz4** | Need quantitative evidence for the two failure modes; unclear mechanism for why they occur; no compute–accuracy analysis; how well I2S improves strategy transfer / what “strategies” mean. | Added a **failure-mode annotation study** on AIME’25 & GPQA for R1-Distill-Qwen-7B and Qwen3-1.7B (Sec. 5.6, Fig. 7, App. D), showing semantic misleading dominates one-shot ICL errors and is reduced by I2S; added a focused **mechanistic discussion** (Sec. 3.2) linking the “distillation demos hurt” paradox to the ICL-as-gradient-descent view and explaining semantic misleading as cross-attention pattern completion and strategy-transfer failure as difficulty in abstracting high-level strategies; introduced **compute–accuracy curves vs. generated tokens** (Sec. 5.5) where I2S lies on a better frontier than CoT / Self-Refine / PHP; clarified that each demo–target pair is a **strategy-transfer setting** and added concrete examples of **transferable insights** (Fig. 3, App. B.3). |
| **u8De** | Similar to Reflect / meta-CoT style prompting; Need explicit efficiency trade-off study; two failure modes not quantified; concerns about prompt sensitivity, context length, and whether insights improve faithfulness. | Framed the work as a new empirical study of a paradoxical regime where **distillation training samples hurt as demos**, and positioned I2S as a design that **forces demos into abstract insights** and decouples answering from raw traces; added accuracy–vs–token curves (Sec. 5.5) showing I2S / I2S+ **match or outperform baselines at the same token budget**; introduced the same **failure-mode statistics** (Sec. 5.6) to show that I2S systematically **reduces semantic-misleading and strategy-transfer errors**; clarified that we use **one fixed, simple prompt set** across all models/datasets without tuning yet still see gains; added Q+A-only ICL experiments (short demos, App. B.1) showing performance still **drops with more demos**, indicating the issue is not only long CoT length. |
| **v1vB** | Focus on one-shot ICL instead of many-shot; quantitative evidences of two observed failure modes; Sec. 4.2 need explicit prompt references; asks why not use self-generated reasoning as demos. | Clarified that for reasoning LMs, "more demos is better" is not guaranteed, and showed empirically that adding more demos can hurt performance for both Q+CoT+A and short Q+A demos (App. B.1); added **GPT-4.1 few-shots experiments** on AIME & GPQA, where I2S 1-shot substantially improves over direct prompting and 2–3 shots give similar or slightly higher gains; supported the diagnosis with the **quantitative failure-mode analysis** (Sec. 5.6), showing that semantic misleading can account for 40–50% of one-shot ICL failures and is consistently **reduced by I2S**; rewrote Sec. 4.2 to **explicitly point to the concrete prompts** and templates; clarified that we deliberately focus on the **teacher-demo setting** that exposes the distillation-demo paradox, and treat self-generated demos as an orthogonal future direction. |
| **J6ds** | Benchmarks and models may be narrow; missing reported standard deviations; minor presentation/typo issues. | Clarified that the existing setup already covers **diverse regimes** (numeric AIME’25, MC GPQA, and two open-ended general reasoning suites) and focused on **deeper analyses** (Sec. 5.5 & 5.6) rather than adding many more tasks; added **Qwen3-14B** experiments showing that I2S/I2S+ also yield **consistent gains** for a larger reasoning model; reported **standard deviations over five runs** in App. B.5; fixed the figure layout and typos noted by the reviewer. |


We understand that this year’s ICLR review process has been especially challenging, and we are sincerely grateful that you took the time to read our lengthy rebuttal and additional experiments. We remain confident that our analysis of why reasoning in-context demonstrations can hurt LRMs and the proposed framework for mitigating these failures at test time will be a valuable contribution to the community!

---

### Meta-Review · Area_Chair_78sg · 2026-01-07

**Summary:**

The work introduces I2S and I2S+ frameworks. They are conceptually simple yet effective, converting demonstrations into reusable reasoning insights and optionally refining reasoning for coherence. The results show consistent accuracy improvements across multiple  reasoning benchmarks. The technical novelty of I2S is limited, which is mainly prompting engineering. The proposed method’s scalability to many-shot settings remains questionable. I2S relies on additional model calls, which thus increases test-time latency.

**Reviewer Concerns:**

The technical novelty of I2S is limited, which is mainly prompting engineering. The proposed method’s scalability to many-shot settings remains questionable. I2S relies on additional model calls, which thus increases test-time latency. These are not well addressed.

**Reviewer Scores:**

reviewers are likely to keep the rating.

---

### Decision · Program_Chairs · 2026-01-26

Reject